# RECAST: Model Reconstruction via Counterfactual-Aware Wasserstein Geometry under Limited Data

Xuan Zhao [* 1]  Lena Krieger [* 1 2]  Zhuo Cao [1]  Arya Bangun [1]  Hanno Scharr [1]  Ira Assent [1 3]

## Abstract

Counterfactual explanations (CFs) help understand machine learning models by identifying minimal input changes that would lead to alternative model outcomes. Recent work demonstrates their utility for reconstructing black-box models, enabling third-party auditing of opaque decision systems for fairness and accountability. Still, CF-based reconstruction may suffer from decision boundary shifts, overfitting, and restrictive assumptions requiring online query access to target platforms. We propose **REconstruction via Counterfactual-Aware waSserstein opTimization (RECAST)** under limited data and restricted access, a behavioral surrogate model based on Wasserstein barycentric prototypes. Our approach addresses decision boundary shifts by incorporating CFs as informative, though less representative, samples for both classes, maintaining high surrogate fidelity in low-sample regimes without requiring online access during reconstruction. To enhance fairness auditing, our method enables systematic group fairness diagnostics. Experiments on real-world datasets and various setups show that **RECAST** effectively achieves high fidelity and query efficiency, as well as stable results even when the access is limited and noisy.

## 1. Introduction

Machine-Learning-as-a-Service (MLaaS) platforms have democratized access to AI systems in high-stakes decision-making systems. Such systems can be audited by reconstruction, i.e., creation of a surrogate model to closely mimic the

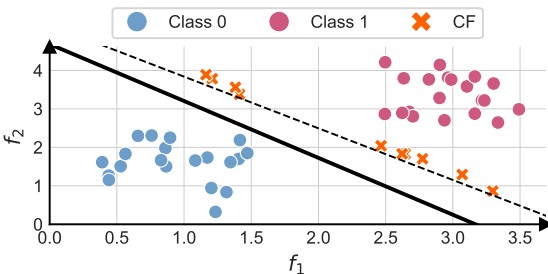

*Figure 1.* CFs (**x**) are generated for class 0 (blue) samples - minimal changes that flip the target's prediction from class 0 → class 1 (pink). Naively treating these as class 1 training samples shifts the surrogate's boundary (solid line) toward class 0, causing class 0 samples to be misclassified as class 1. This is **overconfidence**: the surrogate becomes overly certain about class 1 in regions where the target (dashed line) is not. We instead treat CFs as *less certain* samples, giving them less influence than true class 1 samples.

behaviour of the target model, enabling independent investigation of model behavior, including transparency, accountability, and fairness concerns such as fairwashing (Aïvodji et al., 2021). Reconstruction under limited access is fundamentally a problem of stability: conclusions about model behavior should remain invariant across all models that are behaviorally indistinguishable from the observed data. While studied as model extraction attacks (MEA) (Tramèr et al., 2016; Gong et al., 2020; Carlini et al., 2024; Ferry et al., 2024) in adversarial contexts, reconstruction is crucial to understand model behaviour in high-stakes decision-making systems as required by the EU AI Act[1]. In addition to transparency and accountability, the AI Act also aims to improve diversity and fairness. One way to achieve this is through model reconstruction, by examining how sensitive groups are distributed relative to the learned decision geometry. Recent work leverages counterfactual explanations (CFs), i.e., minimally modified instances that attain a desired prediction, for model reconstruction (Aïvodji et al., 2020; Wang et al., 2022; Dissanayake & Dutta, 2024). This line of work features two main advantages: CFs naturally encode boundary-adjacent information, and improve query efficiency by providing pairs of closely related instances with contrasting predictions.

---

[*]Equal contribution  [1]IAS-8, Forschungszentrum Jülich, Germany  [2]LMU Munich, Munich Center for Machine Learning (MCML), Germany  [3]Department of Computer Science, Aarhus University, Denmark.  Correspondence to: Xuan Zhao <x.zhao@fz-juelich.de>, Lena Krieger <l.krieger@fz-juelich.de>.

*Proceedings of the 43^{rd} International Conference on Machine Learning*, Seoul, South Korea. PMLR 306, 2026. Copyright 2026 by the author(s).

---

[1]artificialintelligenceact.eu/chapter/3

However, existing CF-based reconstruction approaches face critical limitations. First, decision boundary shifts occur, when surrogate models trained on CFs as full class samples (Aïvodji et al., 2020) become overconfident in regions where the CFs are located. Then, the learned decision boundary is shifted incorrectly (Fig. 1), due to margin-based generalization (Shokri et al., 2021), and worsens with one-sided CFs. Two-sided CFs could mitigate this (Wang et al., 2022), but typically platforms do not provide them: rejected loan applicants receive explanations on how to achieve approval, but approved applicants do not learn how their application might have been denied (Aïvodji et al., 2020; Dissanayake & Dutta, 2024). Second, neural network-based methods such as Counterfactual Clamping Attacks (Dissanayake & Dutta, 2024), are prone to overfitting when limited training data is available. As data is usually costly in this setting, constrained by strict query budgets to limit API costs and rate limits, data efficiency is critical. Third, under low-query and one-sided CF settings, decision boundaries are statistically non-identifiable, rendering recovery or approximation of decision boundaries (Dissanayake & Dutta, 2024; Khouna et al., 2025) inapplicable: multiple classifiers may exhibit indistinguishable input–output behavior while having substantially different decision surfaces. Also, data is potentially noisy, as MLaaS platforms may return imperfect or perturbed outputs (Liang et al., 2024). We assume this is a passive observation problem and do not model interactions with a strategic platform using adversarial threat models.

For effective reconstruction of a binary classifier with limited or noisy data, we propose **REconstruction via Counterfactual-Aware waSserstein opTimization (RE-CAST)** which incorporates CFs as *soft* samples for both classes. We leverage Wasserstein barycenters as prototypes for each class to capture the underlying structure of each class distribution in a data-efficient manner, suitable for low-data regimes and noisy samples. To detect asymmetric treatment and decision disparities, we adapt a group fairness diagnostic, based on the reconstructed model, that compares how different sensitive groups (e.g., gender or race) are positioned relative to the barycentric decision geometry.

Our main contributions include:

- Our **Model REconstruction via Counterfactual Constrained Wasserstein Geometry (RECAST)** solution to behavioral reconstruction via counterfactual-aware Wasserstein barycentric prototypes.
- Empirical validation of effectiveness of **RECAST** in various settings, including four datasets, different target model classes, CF generators, and noisy data, demonstrating **high fidelity** and **query efficiency**.
- A geometry-based **fairness diagnostic** to characterize distributional disparities across sensitive groups under distribution shift.

*Table 1.* Model reconstruction using CFs. RECAST intentionally avoids explicit boundary recovery and instead focuses on functionality reconstruction, which is better posed under *one-sided CFs*, *offline access* and *low-query settings*. [*] paper studies only neural networks. [+] requires prior knowledge about the distribution. SAMPLES: Aïvodji et al. (2020).

| Method | One-sided CFs | Model agnostic | Behavior-centric | Access? |
|---|---|---|---|---|
| DualCF | ✗ | ✗ | ✓ | Online |
| TRA | ✓ | ✗ | ✗ | Online |
| CCA | ✓ | ✓ | ✗ | Online |
| SAMPLES | ✓ | ✓[*] | ✓ | Offline[+] |
| RECAST (Ours) | ✓ | ✓ | ✓ | Offline |

## 2. Related Work

Model extraction attacks (MEAs) aim to reconstruct hidden models from queries to an MLaaS platform (Liang et al., 2024; Zhao et al., 2025).

In our work, we focus on functionally equivalent extraction, i.e., recovering a surrogate model whose predictions match those of the target model (Aïvodji et al., 2020) under *offline* access, i.e., based on a fixed sample set constructed without any prior knowledge.

Existing work usually requires *interactive online* query access to the target model *during* the reconstruction process, often not available in practice (Table 1).

Early methods focused on recovering model parameters or explicitly approximating decision boundaries through costly extensive querying (Pal et al., 2020; Jagielski et al., 2020). More recent ones employ CFs, i.e., a minimally modified instance for an input that attains the desired prediction, for instance suggesting an income increase to secure a loan otherwise rejected. Aïvodji et al. (2020), here referred to as SAMPLES, use CFs as *full* labeled data. The approach is vulnerable to decision boundary shift, i.e., it over-confidently assesses samples that lie very close to the decision boundary and are therefore *less* representative (see Fig. 1). In practice, typically only one-sided CFs are available, to support users in understanding negative outcomes, while avoiding information leak about the hidden model.

Recent works, particularly those leveraging CFs, often adopt a boundary-centric perspective. They aim to infer the geometry of the target classifier decision boundary, e.g., with a modified entropy loss that treats CFs as in-between classes (CCA (Dissanayake & Dutta, 2024)) or divide the input space into subspaces and incorporate locally optimal CFs (TRA (Khouna et al., 2025)).

However, the non-identifiability effect (Orekondy et al., 2019; Tramèr et al., 2016) means that under finite samples, different classifiers can exhibit identical or nearly identical classification behavior while having fundamentally different decision boundaries. Another line of work (Aïvodji et al., 2020; Wang et al., 2022) thus emphasizes behavior-centric

reconstruction, to reproduce the input–output behavior of the target model rather than its decision geometry. However, earlier approaches that include CFs, require access to two-sided CFs, e.g., training on pairs, consisting of CFs and CFs for CFs (DualCF (Wang et al., 2022)) or are challenged by decision boundary shifts (SAMPLES (Aïvodji et al., 2020)). We propose a behavior-centric approach, that addresses these limitations, by reconstructing class-level behavior from CFs, which naturally encode boundary-adjacent information, using Wasserstein-based prototypes.

While superficially related to classical learning under data selection bias (Heckman, 1979; Shimodaira, 2000; Zadrozny, 2004), the setting differs in two fundamental aspects. First, the target decision boundary is statistically non-identifiable under offline, one-sided CF access, whereas classical reweighting assumes an identifiable target approximated via ERM on reweighted samples. Second, CFs may overshoot or be systematically biased (see Figure 2), rendering them unreliable as boundary samples. Margin-based methods such as SVMs rely precisely on reliability. CFs instead encode directional distributional constraints between class-conditional distributions, which RECAST handles at distribution level via Wasserstein surrogates rather than via sample-level reweighting or margin maximization.

## 3. The RECAST Method

We start by defining the problem of model reconstruction in low-query regime and with one-sided CF, then describe how our new RECAST approach leverages Wasserstein optimization under CF-consistent uncertainty (Section 3.1) to learn class conditional prototypes (Section 3.2), which we employ in a barycentric prototype classifier (Section 3.3). Finally, we adapt a threshold invariant fairness diagnostic to a Wasserstein geometry perspective (Section 3.4).

**Problem Setting.** We study reconstruction of a binary black-box classifier $m : \mathscr{X} \to \{0, 1\}$, with predictions $m(\boldsymbol{x}) = \mathbb{I}[\hat{y}_m(\boldsymbol{x}) \geq 0.5]$, where $\hat{y}_m$ are (unobserved) prediction scores. Information sources are restricted to:

*Offline collection of labeled inputs*. We are provided with a set of labeled inputs $\{(\boldsymbol{x}, m(\boldsymbol{x}))\}$, that can be grouped by class into $\mathscr{D}_c := \{x : m(\boldsymbol{x}) = c\}$ for $c \in \{0, 1\}$. Our access is restricted to *offline*, i.e., no access to the target classifier *during* reconstruction.

*One-sided CFs.* Additionally, we receive a set of reject-to-accept, i.e., one-sided, CFs $\mathscr{D}_{\mathrm{cf}} := \{\boldsymbol{x}^{\mathrm{cf}}\}$, corresponding to inputs $\boldsymbol{x}$ such that $m(\boldsymbol{x}) = 0$ and $\hat{y}_m(\boldsymbol{x}^{\mathrm{cf}}) \geq 0.5$. One-sided CFs are $\boldsymbol{x}^{\mathrm{cf}} \in \operatorname{argmin}_{\boldsymbol{x}' \in \mathscr{X}} \quad cost(\boldsymbol{x}, \boldsymbol{x}')$ s.t. $\hat{y}_m(\boldsymbol{x}') \geq 0.5$, where $cost(\boldsymbol{x}, \boldsymbol{x}')$ is a user-specified cost measuring the magnitude of the input change, for inputs $\boldsymbol{x}$ with prediction $m(\boldsymbol{x}) = 0$.

Our goal is to reconstruct a surrogate model $\hat{m}$ that is consistent with the information provided by these limited observations, i.e., to *reduce* disagreement rate $\mathrm{Pr}$ under $\mu$:

$$\Pr_{x \sim \mu} \left[ m(x) \neq \hat{m}(x) \right].$$

*Observation* (Non-identifiability). In binary classification with one-sided CF access and limited queries, multiple data-generating distributions can induce identical observable data while differing in their decision boundaries, rendering boundary recovery statistically ill-posed. *Given this non-identifiability, reconstruction quality can only be assessed in terms of behavioral agreement with the target model, rather than recovery of a unique decision boundary.*

Such non-identifiability is well known in classical black-box model reconstruction under finite queries (Tramèr et al., 2016; Orekondy et al., 2019). CFs provide richer information than label queries, but a related ambiguity persists under one-sided CF access, as they do not uniquely constrain the underlying decision surface.

Accordingly, we reconstruct distributional class-level behavior: we aim to reduce disagreement between target model and reconstructed model rather than decision boundaries, and model this behavior through a CF-consistent distributional geometry over probability measures.

Although CFs are defined at the individual level, their impact on model reconstruction is inherently distributional, as they constrain the admissible perturbations between class-conditional data distributions. This naturally motivates a *Wasserstein-geometric* view that measures distributional differences via minimal-cost transport, rather than parametric or pointwise prototype constructions. Prior work has leveraged optimal transport to model counterfactual distributions in a similar distributional setting (You et al., 2025). However, such approaches focus on generating or characterizing CFs themselves, whereas our objective is fundamentally different: we use CFs as structural constraints for reconstructing class-level decision behavior, which uses a Wasserstein geometry over class-conditional measures.

Recent CF-based reconstruction methods (Dissanayake & Dutta, 2024; Khouna et al., 2025) assume that CFs lie near the decision boundary. In practice, however, constraints such as immutable features often force CFs to overshoot the boundary, as illustrated in Figure 2. Since the location of CFs relative to the decision boundary is unknown, they cannot be reliably treated as boundary points. Moreover, when a CF lies on the boundary, assigning it a hard label induces bias (Figure 1). RECAST therefore treats CFs as soft, cross-class constraints, which is less sensitive to the location of the CFs relative to the boundary. Since prototypes aggregate distributional information across all available samples, they average out sample-level noise while preserving the

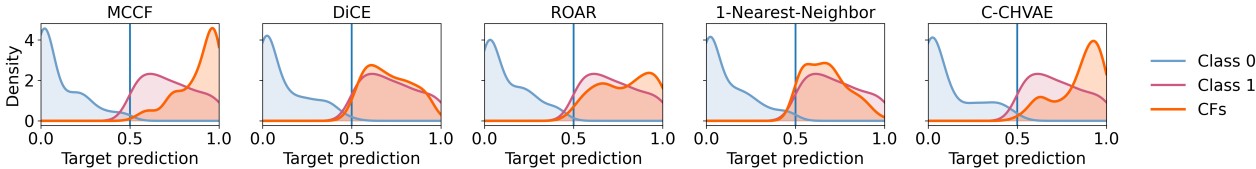

*Figure 2.* Kernel density estimates of target model scores $\hat{y}_m(x)$ for both class inputs, CFs; common CF methods. Concentration well above decision threshold 0.5 means CFs overshoot the boundary, confirming RECAST's approach of avoiding near-boundary assumptions.

structure relevant to the target's decisions. This aggregation effectively increases the usable signal per query, mitigating overfitting in low-query regimes.

### 3.1. CF-consistent Uncertainty

**Definition 3.1** (CF-consistent feasible set). Let $\mathbb{P}_c$ be the empirical class-conditional distributions and $\mathbb{P}_{cf}$ the *one-sided* CF distribution. For each class $c \in \{0, 1\}$, we define

$$\mathscr{C}_c = \Big\{ \mu \in \mathscr{P}_2(\mathscr{X}) : W_2(\mu, \mathbb{P}_c) \le \varepsilon_c, \ W_2(\mu, \mathbb{P}_{cf}) \le \delta_c \Big\},$$

as the *CF-consistent feasible set*, where $\mathscr{P}_2(\mathscr{X})$ denotes the set of probability measures on $\mathscr{X}$ with finite second moments, and $W_2$ denotes the 2-Wasserstein distance (App. A.1). Radii $\varepsilon_c$ and $\delta_c$ encode the uncertainty induced by finite samples, noisy queries, and imperfect CF generation; chosen appropriately, feasible sets $\mathscr{C}_c$ are non-empty. The resulting uncertainty set over joint distributions is

$$\mathscr{U} = \big\{ \mu : \mu(\cdot \mid y = c) \in \mathscr{C}_c, \ c \in \{0, 1\} \big\}.$$

This construction induces a lens-shaped ambiguity set in Wasserstein space (Figure 3). Distributionally Wasserstein robust optimization (WRO) under Wasserstein ambiguity sets is the minimization of the worst case expected loss over all distributions within a Wasserstein ball around an empirical distribution (Mohajerin Esfahani & Kuhn, 2018; Blanchet et al., 2019; Gao & Kleywegt, 2023).

**Relation to Wasserstein robustness.** Classical Wasserstein-robust optimization (WRO) optimizes over a single Wasserstein ball centered at an empirical distribution, with radius that can be statistically calibrated from data. In our setting, however, the ambiguity radii cannot be reliably calibrated from limited one-sided CF data. Thus, our CF-consistent feasible set is the *intersection of two Wasserstein balls*: one centered at the class-conditional data distribution and one centered at the counterfactual distribution, where $B_{W_2}(\mathbb{P}, r) = \{\mu \in \mathscr{P}_2(\mathscr{X}) : W_2(\mu, \mathbb{P}) \le r\}$ denotes the Wasserstein ball of radius $r$ centered at $\mathbb{P}$. This two-anchor construction encodes not only sampling uncertainty but also *geometric constraints induced by CFs*, yielding a *lens-shaped ambiguity set* rather than a single ball. As a result, robustness in our setting is governed by

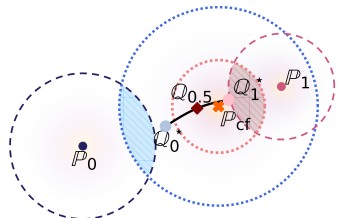

*Figure 3.* **CF-consistent Wasserstein lens.** *All points in the figure correspond to probability distributions in Wasserstein space.* Each class $c$ is associated with a CF-consistent feasible set $\mathscr{C}_c = B_{W_2}(\mathbb{P}_c, \varepsilon_c) \cap B_{W_2}(\mathbb{P}_{cf}, \delta_c)$, given by the intersection (colored face) of a Wasserstein ball centered at $\mathbb{P}_c$ (dashed circle) and a ball centered at $\mathbb{P}_{cf}$ (dotted circle). Although CFs are one-sided, $\mathbb{P}_{cf}$ acts as shared geometric anchor that constrains both classes.

the *relative geometry* between $\mathbb{P}_c$ and $\mathbb{P}_{cf}$, rather than by a single ambiguity radius.

### 3.2. Robust Barycenters

**Theorem 3.2** (Minimizers of Robust Risk Bounds). *Let $(\mathbb{Q}_0, \mathbb{Q}_1) \in \mathscr{P}_2(\mathscr{X})^2$ denote any pair of class-conditional reconstruction distributions, let $\mathscr{U} \subset \mathscr{P}_2(\mathscr{X})^2$ denote the CF-consistent uncertainty set (Definition 3.1), which contains all class conditional data generating distributions consistent with the observed samples and CFs. For any such reconstruction $(\mathbb{Q}_0, \mathbb{Q}_1) \in \mathscr{P}_2(\mathscr{X})^2$, the worst-case distributional reconstruction risk $\mathscr{R}_c(\mathbb{Q}_c) = \sup_{\mu \in \mathscr{C}_c} W_2^2(\mu, \mathbb{Q}_c)$, admits a tractable Wasserstein-robust upper bound. The minimizers are the solutions to the optimization problem*

$$\min_{(\mathbb{Q}_0, \mathbb{Q}_1)} \sum_{c \in \{0, 1\}} \big( (1 - \lambda_c) W_2^2(\mathbb{Q}_c, \mathbb{P}_c) + \lambda_c W_2^2(\mathbb{Q}_c, \mathbb{P}_{cf}) \big).$$

*Minimizing this objective is not equivalent to solving the constrained robust problem directly, but yields a geometry-aware surrogate whose minimizers are stable under CF-consistent perturbations. We measure reconstruction risk in Wasserstein distance to ensure consistency with the geometry of the uncertainty set.*

*Remark* 3.3 (Geometric uniqueness). Without additional regularity assumptions, the minimizer of Theorem 3.2 does not need be unique as a probability measure. Nevertheless, all minimizers lie on the same Wasserstein displacement geodesic between $\mathbb{P}_c$ and $\mathbb{P}_{cf}$, and therefore lead to a unique CF-identifiable Wasserstein distance geometry.

**Barycentric form.** For each $c \in \{0, 1\}$, the optimal $\mathbb{Q}_c^\star$ satisfies

$$\mathbb{Q}_c^\star = \underset{\mu \in P_2(\mathscr{X})}{\operatorname{argmin}} \left( (1 - \lambda_c) W_2^2(\mu, \mathbb{P}_c) + \lambda_c W_2^2(\mu, \mathbb{P}_{\mathrm{cf}}) \right) \quad (1)$$

i.e., $\mathbb{Q}_c^\star$ is the 2-Wasserstein barycenter of $\mathbb{P}_c$ and $\mathbb{P}_{\mathrm{cf}}$ (App. A.2).

**Proof idea.** For each class $c$, a worst-case risk is $\mathscr{R}_c(\mathbb{Q}_c) = \sup_{\mu \in \mathscr{C}_c} W_2^2(\mu, \mathbb{Q}_c)$. Using standard inequalities in Wasserstein space, this worst-case risk admits a robust upper bound by a convex combination of $W_2^2(\mathbb{Q}_c, \mathbb{P}_c)$ and $W_2^2(\mathbb{Q}_c, \mathbb{P}_{\mathrm{cf}})$. Optimizing this bound yields the Wasserstein barycenter in Eq. (1), and separability across classes gives the objective in Theorem 3.2. The full proof is given in App. A.3.

Importantly, our formulation does not require explicit specification of the uncertainty radii $\varepsilon_c$ and $\delta_c$. In our derivation (App. A.3), their relative effect is absorbed into the data-dependent mixing weight $\lambda_c$: selecting an optimal $\lambda_c$ corresponds to selecting optimal radii, and $\lambda_c$ determines the location of the CF-consistent Wasserstein barycenter.

**Canonical instantiation of $\lambda_c$.** To better illustrate the role of $\lambda_c$, we rewrite the class-specific objective (Theorem 3.2) in an equivalent normalized form by dividing by $(1 - \lambda_c)$:

$$\min_{(\mathbb{Q}_0, \mathbb{Q}_1)} \sum_{c \in \{0, 1\}} \left( W_2^2(\mathbb{Q}_c, \mathbb{P}_c) + \frac{\lambda_c}{1 - \lambda_c} W_2^2(\mathbb{Q}_c, \mathbb{P}_{\mathrm{cf}}) \right).$$

This form makes explicit that the influence of CF information is governed by the ratio $\lambda_c/(1 - \lambda_c)$.

Theorem 3.2 yields a family of robust barycenters $\{\mathbb{Q}_c^\star(\lambda_c)\}$. In classical Wasserstein robust optimization, one would select $\lambda_c$ (equivalently, the ambiguity radius) to tighten the bound with respect to the true risk. However, under one-sided CF access and limited data, such statistical calibration is fundamentally infeasible.

In this regime, $\lambda_c$ is not uniquely identifiable from data within the CF-consistent uncertainty set. We consequently instantiate $\lambda_c$ through the observable Wasserstein geometry of the three anchor distributions. Let $A = W_2^2(\mathbb{P}_{\mathrm{cf}}, \mathbb{P}_c)$ and $B = W_2^2(\mathbb{P}_{\mathrm{cf}}, \mathbb{P}_{1-c})$. The relative geometry induces the relation $\frac{\lambda_c}{1 - \lambda_c} = \frac{B}{A + B}$, yielding the closed-form expression $\lambda_c = \frac{B}{A + 2B}$.

From this perspective, the ratio $\lambda_c/(1 - \lambda_c)$ modulates the contribution of one sided CF samples as *soft cross-class* signals: it increases as $\mathbb{P}_{\mathrm{cf}}$ is away from the opposite class $1 - c$, and decreases when it is relatively distant from class $c$. The formulation degrades gracefully at boundary cases: when $\mathbb{P}_{\mathrm{cf}} = \mathbb{P}_1$, the weights reduce to $\lambda_1 = 0$ and $\lambda_0 = 1$, CFs

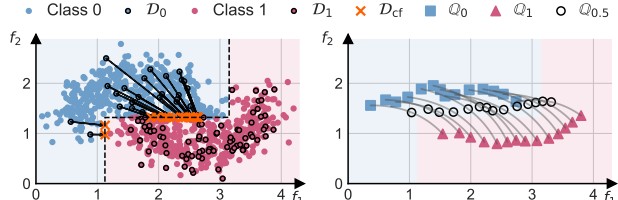

*Figure 4.* **Top**: Full original dataset and decision tree classification (blue and pink); CFs as x (orange); points available for reconstruction circled (black). **Bottom**: Decision region between barycenters: $\mathbb{Q}_0$ (blue), $\mathbb{Q}_1$ (pink); black circles midpoints along the optimal transport path (grey lines), indicating the decision region $\mathbb{Q}_{0.5}$.

carry then no influence on $\mathbb{Q}_0^\star$, and the formulation reduces to standard binary classification. The case $\varepsilon_c = \delta_c = 0$ would require $\mathbb{P}_c = \mathbb{P}_{\mathrm{cf}}$, which contradicts the reject-to-accept CF setting and renders the problem ill-posed. When $\mathbb{P}_{\mathrm{cf}}$ lies approximately equidistant between the two class distributions, the resulting ratio assigns comparable influence to both classes. To set $\lambda_c$ appropriately, we consider the following stability:

**Lemma 3.4** (Stability along the barycentric geodesic). *For class $c \in \{0, 1\}$, let $(Q_c^\star(\lambda))_{\lambda \in [0,1]}$ denote a constant-speed Wasserstein geodesic between $\mathbb{P}_c$ and $\mathbb{P}_{\mathrm{cf}}$. Then, function $\lambda_c \mapsto \mathscr{R}_c(Q_c^\star(\lambda_c))$ is Lipschitz continuous on $[0, 1]$, with Lipschitz constant*

$$L_c = 2 \big( \varepsilon_c + W_2(\mathbb{P}_c, \mathbb{P}_{\mathrm{cf}}) \big) W_2(\mathbb{P}_c, \mathbb{P}_{\mathrm{cf}}).$$

*Consequently, small perturbations of $\lambda_c$ lead to proportionally small changes in the worst-case risk. Proof in App. A.4.*

As a direct consequence, for any (unobservable) optimizer $\lambda_c^\star \in [0, 1]$ of the robust upper bound along the barycentric geodesic, the suboptimality of a chosen $\lambda_c$ is bounded as

$$\mathscr{R}_c(Q_c^\star(\lambda_c)) - \mathscr{R}_c(Q_c^\star(\lambda_c^\star)) \leq L_c |\lambda_c - \lambda_c^\star|.$$

This result ensures that the canonical instantiation remains behaviorally stable even under under small variations of $\lambda_c$.

### 3.3. Barycentric Prototype Classifier

**Definition 3.5** (CF-consistent behavioral equivalence). Two different classifiers $m_1, m_2$ are *behaviorally equivalent* (w.r.t. one-sided CF access), if $\forall \mu \in \mathscr{U}, \Pr_{x \sim \mu}[m_1(x) \neq m_2(x)] = 0$, where the probability denotes the disagreement rate of the two different models under $\mu$.

The CF-consistent uncertainty set $\mathscr{U}$ contains multiple class-conditional distributions that are indistinguishable from the observed data. *Any valid pointwise decision rule must therefore be invariant over all $(\mu_0, \mu_1) \in \mathscr{U}$ of class-conditional data generating distributions, and cannot depend on quantities (such as densities or likelihoods) that vary within this*

*set.* In contrast, Wasserstein distances are invariant to CF-unidentifiable variations, making $W_2(\delta_x, \hat{\mathbb{Q}}_c)$ a natural CF-identifiable score. This is because CF-consistent uncertainty constrains only how probability mass can be transported in the input space, while leaving its local density unspecified; density-based quantities vary under such unidentifiable rearrangements, whereas Wasserstein distances depend solely on the induced transport geometry.

Given the reconstructed class prototypes$(\hat{\mathbb{Q}}_0, \hat{\mathbb{Q}}_1)$, a CF-identifiable decision statistic is given by the 2-Wasserstein distance from a Dirac mass to the class prototypes:

$$\hat{m}(\boldsymbol{x}) := \arg\min_{c \in \{0,1\}} W_2\Big(\delta_{\boldsymbol{x}}, \hat{\mathbb{Q}}_c\Big), \qquad (2)$$

where $\delta_{\boldsymbol{x}}$ denotes the Dirac measure at an input $\boldsymbol{x}$. This classifier should be interpreted as a canonical representative of the CF-consistent behavioral equivalence class, rather than a uniquely identifiable ground-truth decision rule. *Importantly, the proposed framework does not restrict the form of the final decision mechanism, provided it operates solely on CF-identifiable geometric quantities.*

The stability of the barycentric classifier follows directly from the Wasserstein geometry. By Corollary A.4 in App. A.3, the induced pointwise decision score is Lipschitz continuous with respect to perturbations of the prototype distributions under $W_2$. This stability, combined with the distributional smoothing induced by the barycentric aggregation, yields a uniform bound on the prediction disagreement between the classifiers. Thus, for any $\mathbb{Q}'_c$ distribution in CF-consistent feasible set $\mathscr{C}_c$: $W_2(\mathbb{Q}_c, \mathbb{Q}'_c) \leq \sqrt{\mathscr{R}_c(\mathbb{Q}_c)}$, and small reconstruction risks $\mathscr{R}_c(\mathbb{Q}_c)$ imply small variations in the classification behavior across all CF-consistent realizations.

This classification rule induces a decision region in Wasserstein space. Let $T$ denote an optimal transport map from $\hat{\mathbb{Q}}_0$ to $\hat{\mathbb{Q}}_1$. The Wasserstein geodesic between the two class prototypes is given by the displacement interpolation $\mathbb{Q}_\gamma := \big((1-\gamma)\mathbf{Id} + \gamma T\big)_{\#} \hat{\mathbb{Q}}_0$, $\gamma \in [0,1]$. The decision region is characterized by the midpoint of this geodesic, $\mathbb{Q}_{0.5} = \big(\frac{1}{2}\mathbf{Id} + \frac{1}{2}T\big)_{\#} \hat{\mathbb{Q}}_0$, which represents the distribution equidistant from both class prototypes in Wasserstein distance (Figure 4). In this geometric view, $\hat{\mathbb{Q}}_0$ and $\hat{\mathbb{Q}}_1$ act as distributional *prototypes*, summarizing the CF-consistent class-conditional geometry. Classification is performed by comparing the Wasserstein distance of a point to these prototypes, yielding a prototype-based classifier in Wasserstein space. For the overall geometric relationship between CF-consistent lens and barycentric prototypes see Figure 3.

### 3.4. Threshold-invariant Fairness Diagnostic

RECAST also supports fairness auditing by adapting a threshold-invariant fairness diagnostic in the perspective of Wasserstein geometry. Given the barycentric class prototypes $(Q_0^\star, Q_1^\star)$, we define a Wasserstein-geometric score $p : \mathscr{X} \to (0,1)$ as:

$$p(\boldsymbol{x}) = \sigma\Big(W_2(\delta_{\boldsymbol{x}}, Q_0^\star) - W_2(\delta_{\boldsymbol{x}}, Q_1^\star)\Big)$$

where $\sigma$ denotes the sigmoid function. Based on Chen & Wu (2020), we adapt Threshold Invariant Demographic Parity (TIDP) and Threshold Invariant Equalized Odds (TIEO), i.e., equal selection rates and equal prediction accuracy across sensitive attributes independent of a decision threshold, in the perspective of Wasserstein geometry. In contrast to previous work, our score operates on entire score distributions, capturing disparities that persist across all decision thresholds. We quantify group-level disparities by comparing the distributions of this score across sensitive groups that share a sensitive attribute $S \in \{0,1\}$. We compute the Wasserstein-1 distance between the score distributions conditioned on different values of $S$ to assess TIDP, and between score distributions conditioned on both $S$ and the label $Y \in \{0,1\}$ to assess TIEO. As the scores are induced by barycentric prototypes defined over the CF-consistent feasible sets, the resulting fairness diagnostics are evaluated with respect to a shared uncertainty-aware decision geometry and quantify how different groups are positioned relative to the barycentric decision geodesic. Equations in App. A.7.

**Summary.** Under one-sided CF access the target decision boundary is fundamentally non-identifiable, so reconstruction cannot recover a unique ground-truth classifier. RECAST therefore (i) constructs a CF-consistent Wasserstein uncertainty set, (ii) computes $\hat{\mathbb{Q}}_c$ as Wasserstein barycenters, and (iii) predicts with the classification rule, Eq. (2). Minimizing the resulting Wasserstein reconstruction risk promotes stability of decision behavior under CF-consistent perturbations, yielding reduced prediction disagreement.

## 4. Experiments

We evaluate RECAST across four real-world datasets, three target model families, and multiple CF generation methods, assessing fidelity, robustness, and fairness preservation.

### 4.1. Experimental Setup

**Data.** We study four publicly available binary classification datasets, that are often used in literature and cover a variety of properties, e.g., dimensionalities, dataset size, and tasks: Adult Income (Becker, 1996), COMPAS (Angwin et al., 2016), HELOC (FICO, 2018), and California Housing (Pace & Barry, 1997). We include exploratory cases on additional data modalities in App. C.6 and C.7.

**Baselines.** To the best of our knowledge, there is limited prior work on reconstruction under restricted data access

and only one-sided CFs (Table 1). As such, we adapt three representative baselines to *offline-only*, i.e., no querying during reconstruction. Aïvodji et al. (2020), referred to as *SAMPLES*, incorporates CFs as ordinary samples. *CCA*, Counterfactual Clamping Attack, modifies entropy loss to explicitly incorporate CFs (Dissanayake & Dutta, 2024). TRA (Khouna et al., 2025) assumes prior knowledge that the target classifier is a decision tree with axis-parallel splits, and is therefore included only when such structural information is available; in contrast, RECAST and the other baselines operate without access to this form of model-specific prior. For all baselines, we use hyperparameters recommended in the original papers.

**Barycentric prototype computation.** We compute the barycenters $\mathbb{Q}_c$ using optimization Algorithm 1 in App. A. We optimize an entropically regularized (Sinkhorn-smoothed) surrogate of the Wasserstein barycenter objective for computational efficiency, following standard practice in optimal transport (Chizat et al., 2020), theoretical guarantees and an ablation regarding the blur parameter are included in App. A.5 and A.6. All solver settings and hyperparameters are kept fixed across experiments. All implementation details, including initialization, solver settings, and hyperparameters also in App. B. Code is available online.[2]

**Reconstruction.** As target models, we study *neural networks (MLP), logistic regression (LR), and tree-based classifiers (DT)* trained on 80% of the original datasets, which remains unknown during the reconstruction phase. In each experiment, we randomly sample a certain query size, e.g., 100 instances from class 0 and 100 from class 1, as $\mathscr{D}_0$, $\mathscr{D}_1$, resp. From class 0 samples various CF methods are used to generate CFs $\mathscr{D}_{cf}$. We evaluate the reconstructed models on a held-out reference set $\mathscr{D}_{ref}$, disjoint from the reconstruction data.

We use *fidelity* (Aïvodji et al., 2020) rather than accuracy as our primary metric: a surrogate that faithfully mimics the target model's decision, including its systematic errors, should score highly regardless of ground-truth label agreement, since behavioral reconstruction, not predictive performance is the relevant objective for reconstruction (see App. C.3 for a joint comparison).

Fidelity between the target model $m$ and the surrogate $\hat{m}$ on a reference set $\mathscr{D}_{ref}$ is defined as:

$$\text{Fid}_{m,\mathscr{D}_{ref}}(\hat{m}) = \frac{1}{|\mathscr{D}_{ref}|} \sum_{\boldsymbol{x} \in \mathscr{D}_{ref}} \mathbb{I}_{[0,1]} \left[ m(\boldsymbol{x}) = \hat{m}(\boldsymbol{x}) \right].$$

We repeat each experiment ten times with different random seeds and report the mean and variance of our method. In

---

[2] https://github.com/zhaoxuan00707/ce_reconstruction

addition to the balanced setting, we conduct supplementary experiments with imbalanced $\mathscr{D}_0$ and $\mathscr{D}_1$; see results in App. B.

**CF Generation.** We evaluate several CF generation methods, including MCCF (Wachter et al., 2017), which seeks CFs with minimal input changes. In our implementation, we add an $\ell_1$ regularization term to encourage sparse input changes. DiCE (Mothilal et al., 2020) ensures actionability by enforcing immutable features. To improve robustness under model shifts, we adopt ROAR (Upadhyay et al., 2021). For realism, we apply 1-Nearest-Neighbor from the desired class and C-CHVAE (Pawelczyk et al., 2020), which uses variational autoencoders. Details in App. B.

**Robustness Evaluation Protocol.** Since RECAST is derived under CF-consistent uncertainty, we evaluate not only average fidelity under i.i.d. test data, but also robustness under distributional shifts. Although our uncertainty is specified over training distributions, its effect manifests as sensitivity to distributional shifts during test. We thus evaluate robustness by probing fidelity under perturbations of the test distribution, which operationalizes different plausible realizations of the underlying $\mathbb{Q}_c$. Specifically, for each reference test set $\mathscr{D}_{ref}$, we construct a family of shifted test sets $\mathscr{D}_{ref} = \{\boldsymbol{x} + \epsilon : \boldsymbol{x} \in \mathscr{D}_{ref}, \ \epsilon \sim \mathscr{N}(0, \tau^2 I)\}$, with noise levels $\tau \in \{0.05, 0.1, 0.2, 0.4\}$.

Although average fidelity is a widely used metric for model reconstruction, it can be dominated by samples far from the decision threshold, whose predictions are inherently stable under perturbations. Consequently, high overall fidelity may mask disagreement in decision-sensitive regions. We thus also evaluate robustness on a near-threshold subset where the target model likely exhibits higher uncertainty. Although the true decision boundary is not identifiable, $|\hat{y}_m(\boldsymbol{x}) - 0.5| \leq \gamma$ captures inputs most susceptible to distributional perturbations (Blanchet et al., 2024). We define the near-threshold set as $\mathscr{D}_{near} = \{\boldsymbol{x} \in \mathscr{D}_{ref} : |\hat{y}_m(\boldsymbol{x}) - 0.5| \leq \gamma\}$, with $\gamma = 0.05$. We report fidelity on both the full set $\mathscr{D}_{ref}$ and on $\mathscr{D}_{near}$. Note that access to the target model's continuous output scores is used only during the evaluation phase to construct diagnostic subsets $\mathscr{D}_{near}$, and is neither available nor used during the reconstruction phase. We additionally evaluate robustness under cross-domain covariate shift, details of that protocol are in App. C.4.

**Fairness Diagnostic Protocol.** For each distribution-shift level $\tau$, we report the absolute difference between the fairness diagnostics for target model $m$ and reconstructed model $\hat{m}$ over the corresponding shifted reference set $\mathscr{D}_{ref}$. We show how well the reconstructed models preserve the original decision disparities, and assess their use for auditing.

*Table 2.* Fidelity. Multilayer perceptron (MLP), logistic regression (LR), decision tree (DT) target models, low-query size 100. RECAST consistently achieves higher (in one case comparable) fidelity across datasets and target models. TRA only feasible with DT target.

| Dataset | MLP (Target Model) | | | LR (Target Model) | | | DT (Target Model) | | | |
|---|---|---|---|---|---|---|---|---|---|---|
| | SAMPLES | CCA | RECAST (Ours) | SAMPLES | CCA | RECAST (Ours) | SAMPLES | CCA | RECAST (Ours) | TRA |
| Adult In. | $0.801 \pm 0.035$ | $0.843 \pm 0.020$ | $\mathbf{0.908 \pm 0.034}$ | $0.909 \pm 0.008$ | $0.816 \pm 0.025$ | $\mathbf{0.913 \pm 0.033}$ | $0.819 \pm 0.017$ | $0.827 \pm 0.018$ | $\mathbf{0.894 \pm 0.040}$ | $0.842 \pm 0.020$ |
| COMPAS | $0.415 \pm 0.060$ | $0.765 \pm 0.020$ | $\mathbf{0.855 \pm 0.022}$ | $0.543 \pm 0.004$ | $0.785 \pm 0.026$ | $\mathbf{0.821 \pm 0.047}$ | $0.396 \pm 0.003$ | $0.657 \pm 0.027$ | $\mathbf{0.869 \pm 0.049}$ | $0.826 \pm 0.007$ |
| HELOC | $0.427 \pm 0.121$ | $0.644 \pm 0.057$ | $\mathbf{0.696 \pm 0.078}$ | $0.551 \pm 0.013$ | $0.670 \pm 0.013$ | $\mathbf{0.798 \pm 0.025}$ | $0.620 \pm 0.008$ | $0.702 \pm 0.023$ | $\mathbf{0.772 \pm 0.029}$ | $0.702 \pm 0.024$ |
| Housing | $0.459 \pm 0.102$ | $\mathbf{0.716 \pm 0.070}$ | $0.712 \pm 0.121$ | $0.477 \pm 0.001$ | $0.656 \pm 0.014$ | $\mathbf{0.789 \pm 0.094}$ | $0.554 \pm 0.012$ | $0.619 \pm 0.034$ | $\mathbf{0.774 \pm 0.149}$ | $0.654 \pm 0.015$ |

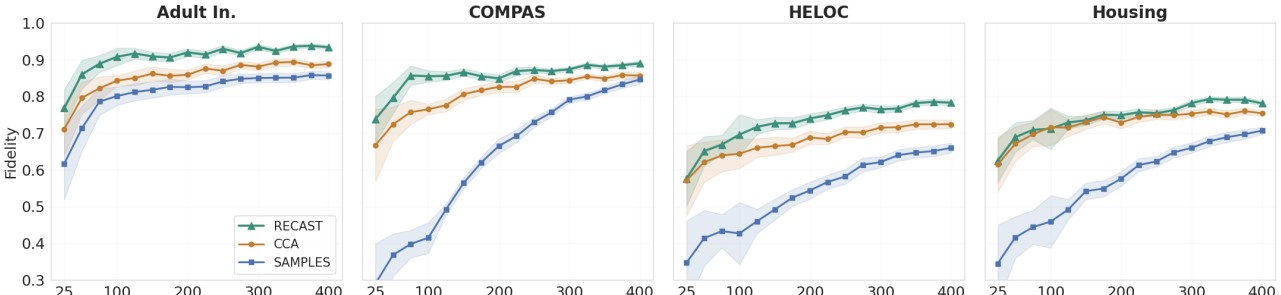

*Figure 5.* Fidelity (mean $\pm$ std) of surrogate models on real-world datasets under varying query sizes. RECAST with superior performance.

**Ablation Study.** We conduct an ablation study on different prototype constructions, distance metrics, and CFs integration strategies in RECAST with all details in App. C.1.

### 4.2. Reconstruction Fidelity

Table 2 summarizes *fidelity* results across target classifiers, with query size 100 and MCCF as CF generation method. In almost all cases, our method achieves superior fidelity to *SAMPLES* and *CCA*, comparable in one. We also study how the amount of available training data influences the performance of the reconstructed models with query sizes from 25 to 400 in Figure 5. Notably, our approach demonstrates a clear advantage when the query size is small, highlighting its efficiency in low-query regimes.

### 4.3. Effect of CF Geometry

RECAST consistently achieves high fidelity across CF generation strategies, see Table 3. Baseline methods exhibit substantially larger fluctuations, indicating sensitivity to the properties of CFs. Unlike baselines, RECAST neither interprets CFs as true samples (SAMPLES) nor assumes proximity to the decision boundary (CCA), properties that depend on the generator, illustrated in Figure 2.

### 4.4. Robustness under Perturbations

We study robustness along two axes: *behavioral stability* under additive perturbations of the test distribution and *generalization* under realistic distribution shift across domains. Both test whether RECAST's behavioral agreement with the target model holds beyond the reconstruction setting. Disagreement between a reconstructed surrogate and the target

model may arise when perturbations cause inputs to cross decision surfaces, an effect most pronounced near the decision threshold. As the perturbation strength increases, fidelity decreases for all methods (see Figure 6). Still, RECAST degrades substantially more gracefully than the baselines, and SAMPLES and CCA suffer pronounced performance drops even under moderate noise. This gap is even more pronounced in the near-threshold region $\mathscr{D}_{\mathrm{near}}$ most sensitive to perturbations.

Along the second axis, we evaluate robustness under realistic distribution shift using Folktables ACSIncome dataset (Ding et al., 2021), training on California 2018 data and evaluating on Michigan 2014, a setting that combines both geographical and temporal variation (see App. C.4, Table 7). RECAST consistently achieves highest fidelity across query budgets (100 and 150), while SAMPLES leads on accuracy but lags notably on fidelity, consistent with the distinction drawn above, where accuracy and fidelity capture fundamentally different objectives. These results confirm that RECAST maintains strong behavioral consistency with the target model even under substantial distribution shifts.

*Table 3.* Fidelity, varying counterfactual generation methods on Adult, query size 100, target MLP. RECAST shows robustness with high fidelity across diverse CF generators.

| CF Generation | Baselines | | |
|---|---|---|---|
| | SAMPLES | CCA | RECAST (Ours) |
| MCCF | 0.793 | 0.893 | **0.913** |
| DiCE | **0.933** | 0.903 | 0.923 |
| ROAR | 0.735 | 0.735 | **0.910** |
| 1-Nearest-Neighbor | 0.712 | 0.835 | **0.916** |
| C-CHVAE | 0.357 | 0.817 | **0.923** |

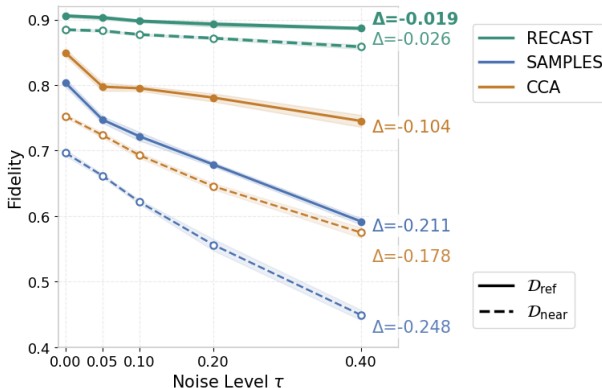

*Figure 6.* Robustness to additive noise on Adult. Fidelity as a function of the noise level $\tau$ for both full reference set $\mathscr{D}_{\text{ref}}$ and near-threshold subset $\mathscr{D}_{\text{near}}$, which captures decision-sensitive inputs. $\Delta$ denotes the fidelity drop between $\tau = 0$ and $\tau = 0.4$.

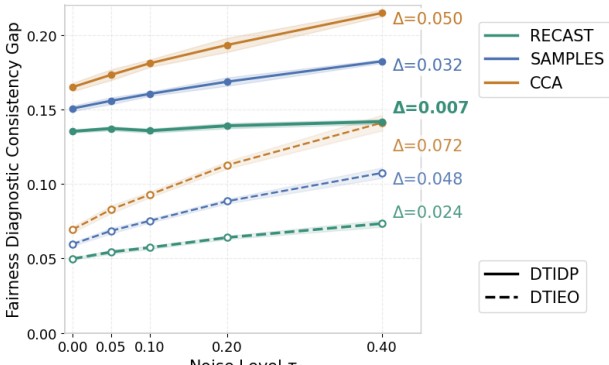

*Figure 7.* Absolute fairness (DTIDP (solid) and DTIEO (dashed)) differences between reconstructed and target model as a function of noise level $\tau$. Lower values mean more faithful preservation of target model fairness characteristics.

### 4.5. Fairness Diagnostics

*The goal of the proposed fairness diagnostics is to assess whether a reconstructed surrogate preserves the fairness-related behavior of the target model, under CF-consistent distributional uncertainty.* These diagnostics are thus intended solely for auditing purposes and do not constitute fairness performance or debiasing claims. Figure 7 reports the fairness diagnostic error between the reconstructed surrogate and the target model under increasing perturbations. Lower values indicate that the surrogate more faithfully preserves the fairness characteristics of the target model. RECAST consistently exhibits smaller diagnostic consistency gap and more graceful degradation with noise.

### 4.6. Effect of Query Budget

RECAST is designed for limited query access and one-sided CFs, where direct reconstruction of the target model is fundamentally sample-inefficient and the decision boundary is not statistically identifiable. To make this regime explicit, we additionally study how reconstruction performance varies with available query budget.

Specifically, we compare RECAST with a no CFs baseline over number of queried labeled instances, and identify the crossover point at which direct supervised reconstruction becomes competitive (Figure 9, App. C.5). This analysis reveals a clear practical boundary: in low-query regimes, CFs provide crucial decision-relevant information that cannot be recovered from samples alone, enabling RECAST to substantially outperform both CF-based and non-CF baselines. As the query budget increases and empirical samples begin to adequately cover decision-relevant regions, the advantage of CF-based reconstruction naturally diminishes, and simpler supervised models become sufficient. Note that with sufficient data, the Wasserstein mixing weights could be

calibrated via classical WRO methods, but this lies beyond the scope of the low-query regime studied here.

In privacy-sensitive auditing contexts, where datasets are small, access is restricted, and queries are costly, such high-capacity approaches become infeasible, making lightweight, geometry-driven reconstruction methods such as RECAST particularly relevant in practice.

We acknowledge that Wasserstein distance can be computationally demanding in large-scale settings. However, in the low-data regimes considered here, this cost is not a practical bottleneck. In runtime experiments (App. C.8), we observe training times near linear in query size.

## 5. Conclusion and Future Work

We propose RECAST, a model reconstruction approach grounded in Wasserstein geometry that captures relationships between data distributions, enabling the formation of robust class prototypes that effectively represent both labeled data and CFs, under one-sided CF access and limited query budgets.

In our experiments, we demonstrate that RECAST maintains high fidelity in small data regimes, where overfitting and poor generalization are prevalent concerns. Importantly, RECAST is not designed to compete with large-query model extraction: it is designed for reconstruction scenarios where data is scarce, access is restricted, and CFs are the only boundary-relevant signal.

Our approach is currently limited to binary classification, which is common in literature since most applications consider accept-reject decision. Extending RECAST to multi-class settings, and leveraging prior knowledge constitute interesting directions for future work.

## Acknowledgements

This work was partially funded by project W2/W3-108 Initiative and Networking Fund of the Helmholtz Association. We gratefully acknowledge the computing time granted through project XAI (No. 65881) on the supercomputer JURECA at Jülich Supercomputing Centre (JSC).

## Impact Statement

This work studies classifier reconstruction under restricted data access and one-sided CF supervision, a setting that arises naturally when interacting with black-box or proprietary decision systems. By characterizing what aspects of model behavior are fundamentally identifiable under such constraints, our framework enables high-fidelity reconstruction of decision behavior, supporting the auditing and investigation of deployed models for fairness, bias, and accountability concerns in settings with limited model access. Importantly, our approach does not aim to recover a unique ground-truth classifier or individual training examples, but instead focuses on CF-identifiable behavioral structure derived from distributional and counterfactual information.

The primary impact of our work involves reducing the barriers to model reconstruction. Organizations and communities affected by ML decision-making systems, e.g., loan applications, hiring, or healthcare, often lack resources to audit the systems they rely on. Our approach offers these groups a practical tool to investigate issues, such as disparate impact, discrimination, or violations of fairness principles in deployed models, even when model owners are unwilling or unable to grant direct access.

Our work supports the regulatory landscape around responsible, transparent and accountable AI, as policymakers worldwide establish requirements for algorithmic impact assessments and accountability.

While our method does introduce security considerations for information leakage, the societal benefit of enabling fairness investigations outweighs these concerns in contexts involving high-stakes decisions affecting vulnerable populations.

Our work is likely to advance the field toward the establishment of more trustworthy and responsible AI systems by improving accountability.

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

# A. Wasserstein Distance and Barycenters

## A.1. Wasserstein Distance

We briefly recall the notions from optimal transport that are required for the formulation and analysis of our reconstruction framework, in particular Definition 3.1 and Theorem 3.2. Let $\mathscr{P}_2(\mathbb{R}^d)$ denote the set of probability measures on $\mathbb{R}^d$ with finite second moments.

**2-Wasserstein distance.** For $\mu, \nu \in \mathscr{P}_2(\mathbb{R}^d)$, the squared 2-Wasserstein distance is defined as

$$W_2^2(\mu, \nu) := \inf_{\gamma \in \Pi(\mu,\nu)} \int_{\mathbb{R}^d \times \mathbb{R}^d} \|x_1 - x_2\|_2^2 \, d\gamma(x_1, x_2), \tag{3}$$

where $\Pi(\mu, \nu)$ denotes the set of couplings (transport plans) with marginals $\mu$ and $\nu$.

The Wasserstein distance induces a geometry on probability measures that accounts for both mass displacement and spatial structure, and is therefore well suited for comparing distributions arising from localized perturbations such as counterfactual examples.

**Monge formulation.** When there exists a measurable transport map $T : \mathbb{R}^d \to \mathbb{R}^d$ such that $T_{\#}\mu = \nu$, the optimal transport problem admits the equivalent Monge formulation

$$T = \arg \min_{T_{\#}\mu=\nu} \int_{\mathbb{R}^d} \|x - T(x)\|_2^2 \, d\mu(x). \tag{4}$$

In general, such a map need not exist; however, the Kantorovich formulation in (3) always remains well defined and is the form used throughout this work.

## A.2. Wasserstein Barycenter

Given a finite collection of probability measures $\{\mu_i\}_{i=1}^N \subset \mathscr{P}_2(\mathbb{R}^d)$ and nonnegative weights $\{\lambda_i\}_{i=1}^N$ satisfying $\sum_{i=1}^N \lambda_i = 1$, the Wasserstein barycenter is defined as

$$\mu^* = \arg \min_{\mu \in \mathscr{P}_2(\mathbb{R}^d)} \sum_{i=1}^N \lambda_i W_2^2(\mu, \mu_i). \tag{5}$$

The barycenter $\mu^*$ provides a distributional prototype that summarizes the input measures in Wasserstein space. Unlike Euclidean averaging, this formulation aligns probability mass prior to aggregation, enabling the resulting prototype to reflect decision-relevant geometric structure induced by the target classifier.

In our setting, the input measures correspond to distributions derived from counterfactual perturbations. These perturbations encode geometric information about the decision behavior of the black-box model, allowing the barycenter to serve as a behaviorally meaningful class-level representation without requiring explicit recovery of the decision boundary.

### A.2.1. RELEVANT PROPERTIES

We summarize only the properties required for the theoretical results in Section 3.

- **Existence.** If each $\mu_i \in \mathscr{P}_2(\mathbb{R}^d)$, then a Wasserstein barycenter $\mu^*$ exists.

- **Convexity of the objective.** The barycenter objective in Eq. (5) is convex in $\mu$, ensuring that all minimizers attain the same optimal value.

- **Non-uniqueness.** The barycenter may not be unique; however, as shown in Theorem 3.2, this non-identifiability does not affect the induced decision rule, which depends only on Wasserstein distance values rather than on a particular minimizer.

## A.3. Robust Risk Bounds and Barycentric Optima

We begin by proving Theorem 3.2, which establishes that the minimizers of a CF-consistent robust risk upper bound admit a Wasserstein barycentric form. The subsequent results in this subsection further clarify the geometric interpretation and stability properties of these barycentric optima.

**Proof of Theorem 3.2.**

*Proof.* Fix a class $c \in \{0, 1\}$. We derive a tractable upper bound on the CF-consistent worst-case risk

$$\mathscr{R}_c(\mathbb{Q}_c) := \sup_{\mu \in \mathscr{C}_c} W_2^2(\mu, \mathbb{Q}_c), \qquad \mathscr{C}_c = \{\mu : \ W_2(\mu, \mathbb{P}_c) \le \varepsilon_c, \ W_2(\mu, \mathbb{P}_{cf}) \le \delta_c\}.$$

For any $\mu \in \mathscr{C}_c$, the triangle inequality yields

$$W_2(\mu, \mathbb{Q}_c) \le W_2(\mathbb{Q}_c, \mathbb{P}_c) + \varepsilon_c, \qquad W_2(\mu, \mathbb{Q}_c) \le W_2(\mathbb{Q}_c, \mathbb{P}_{cf}) + \delta_c.$$

Consequently,

$$W_2^2(\mu, \mathbb{Q}_c) \le \min\Big\{\big(W_2(\mathbb{Q}_c, \mathbb{P}_c) + \varepsilon_c\big)^2, \ \big(W_2(\mathbb{Q}_c, \mathbb{P}_{cf}) + \delta_c\big)^2\Big\}.$$

Since for any $\alpha_c \in [0, 1]$ and any $a, b \ge 0$, $\min\{a, b\} \le (1 - \alpha_c)a + \alpha_c b$, we obtain

$$W_2^2(\mu, \mathbb{Q}_c) \le (1 - \alpha_c)\big(W_2(\mathbb{Q}_c, \mathbb{P}_c) + \varepsilon_c\big)^2 + \alpha_c\big(W_2(\mathbb{Q}_c, \mathbb{P}_{cf}) + \delta_c\big)^2.$$

To control the cross terms, we apply Young's inequality: for any $\eta_1, \eta_2 > 0$ and any $a \ge 0$,

$$2\varepsilon_c a \le \eta_1 a^2 + \frac{\varepsilon_c^2}{\eta_1}, \qquad 2\delta_c a \le \eta_2 a^2 + \frac{\delta_c^2}{\eta_2}.$$

Applying this inequality to the two squared terms yields

$$\big(W_2(\mathbb{Q}_c, \mathbb{P}_c) + \varepsilon_c\big)^2 \le (1 + \eta_1)W_2^2(\mathbb{Q}_c, \mathbb{P}_c) + \Big(1 + \frac{1}{\eta_1}\Big)\varepsilon_c^2,$$

$$\big(W_2(\mathbb{Q}_c, \mathbb{P}_{cf}) + \delta_c\big)^2 \le (1 + \eta_2)W_2^2(\mathbb{Q}_c, \mathbb{P}_{cf}) + \Big(1 + \frac{1}{\eta_2}\Big)\delta_c^2.$$

Combining the above bounds, we obtain for all $\mu \in \mathscr{C}_c$,

$$W_2^2(\mu, \mathbb{Q}_c) \le (1 - \alpha_c)(1 + \eta_1)W_2^2(\mathbb{Q}_c, \mathbb{P}_c) + \alpha_c(1 + \eta_2)W_2^2(\mathbb{Q}_c, \mathbb{P}_{cf}) + K_c,$$

where

$$K_c = (1 - \alpha_c)\Big(1 + \frac{1}{\eta_1}\Big)\varepsilon_c^2 + \alpha_c\Big(1 + \frac{1}{\eta_2}\Big)\delta_c^2$$

is independent of $\mathbb{Q}_c$.

Taking the supremum over $\mu \in \mathscr{C}_c$ yields the robust upper bound

$$\mathscr{R}_c(\mathbb{Q}_c) \le (1 - \alpha_c)(1 + \eta_1)W_2^2(\mathbb{Q}_c, \mathbb{P}_c) + \alpha_c(1 + \eta_2)W_2^2(\mathbb{Q}_c, \mathbb{P}_{cf}) + K_c.$$

Since $K_c$ does not depend on $\mathbb{Q}_c$, minimizing this bound is equivalent to minimizing its quadratic part. Defining

$$\lambda_c = \frac{\alpha_c(1 + \eta_2)}{(1 - \alpha_c)(1 + \eta_1) + \alpha_c(1 + \eta_2)} \in [0, 1],$$

we obtain the barycentric objective

$$(1 - \lambda_c)W_2^2(\mathbb{Q}_c, \mathbb{P}_c) + \lambda_c W_2^2(\mathbb{Q}_c, \mathbb{P}_{cf}),$$

whose minimizer is the 2-Wasserstein barycenter of $\mathbb{P}_c$ and $\mathbb{P}_{cf}$ with weights $(1 - \lambda_c, \lambda_c)$. $\square$

**Existence and interpretation of barycentric optima.** Theorem 3.2 shows that minimizing a suitable robust upper bound on the CF-consistent worst-case risk leads to a Wasserstein barycentric prototype. We formalize this consequence below and clarify the role of the associated bounding parameters.

**Lemma A.1** (Barycentric optima induced by robust bounds). *For each class* $c \in \{0, 1\}$, *consider the CF-consistent worst-case risk*

$$\mathscr{R}_c(\mathbb{Q}_c) := \sup_{\mu \in \mathscr{C}_c} W_2^2(\mu, \mathbb{Q}_c), \qquad \mathscr{C}_c = \{\mu : W_2(\mu, \mathbb{P}_c) \leq \varepsilon_c, \ W_2(\mu, \mathbb{P}_{cf}) \leq \delta_c\}.$$

*There exist parameters* $\alpha_c \in [0, 1]$ *and* $\eta_1, \eta_2 > 0$ *such that the minimizer of a valid robust upper bound on* $\mathscr{R}_c(\mathbb{Q}_c)$ *is given by the* 2-*Wasserstein barycenter between* $\mathbb{P}_c$ *and* $\mathbb{P}_{cf}$ *with weight*

$$\lambda_c = \frac{\alpha_c(1 + \eta_2)}{(1 - \alpha_c)(1 + \eta_1) + \alpha_c(1 + \eta_2)}.$$

*Equivalently, for this choice of parameters,*

$$\arg \min_{\mathbb{Q}_c} \left\{ (1 - \lambda_c) W_2^2(\mathbb{Q}_c, \mathbb{P}_c) + \lambda_c W_2^2(\mathbb{Q}_c, \mathbb{P}_{cf}) \right\}$$

*minimizes a robust upper bound on the CF-consistent worst-case risk* $\mathscr{R}_c(\mathbb{Q}_c)$.

*Proof.* This result follows directly from the proof of Theorem 3.2. For any $\alpha_c \in [0, 1]$ and $\eta_1, \eta_2 > 0$, the derivation above yields a robust upper bound of the form

$$\mathscr{R}_c(\mathbb{Q}_c) \leq (1 - \alpha_c)(1 + \eta_1) W_2^2(\mathbb{Q}_c, \mathbb{P}_c) + \alpha_c(1 + \eta_2) W_2^2(\mathbb{Q}_c, \mathbb{P}_{cf}) + K_c,$$

where $K_c$ is independent of $\mathbb{Q}_c$.

Since multiplication of the objective by a positive constant does not affect its minimizers, the quadratic part of the bound can be renormalized to yield the barycentric objective with weight

$$\lambda_c = \frac{\alpha_c(1 + \eta_2)}{(1 - \alpha_c)(1 + \eta_1) + \alpha_c(1 + \eta_2)}.$$

$\square$

*Remark* A.2 (Barycentric lens of CF-consistent optima). For any fixed $\eta_1, \eta_2 > 0$, the mapping

$$\alpha_c \longmapsto \lambda_c = \frac{\alpha_c(1 + \eta_2)}{(1 - \alpha_c)(1 + \eta_1) + \alpha_c(1 + \eta_2)}$$

is continuous and strictly increasing from $[0, 1]$ to $[0, 1]$. Consequently, by varying $\alpha_c$, the induced barycentric weight $\lambda_c$ ranges over the entire unit interval. Thus, every Wasserstein barycenter along the geodesic between $\mathbb{P}_c$ and $\mathbb{P}_{cf}$ is the minimizer of some CF-consistent robust upper bound on $\mathscr{R}_c$.

Geometrically, the family of robust optima fills the Wasserstein "lens" induced by the intersection $B_{W_2}(\mathbb{P}_c, \varepsilon_c) \cap B_{W_2}(\mathbb{P}_{cf}, \delta_c)$, justifying the barycentric prototype interpretation used in Figure 3.

*Remark* A.3 (Why no explicit bound tuning is required). Different choices of the bounding parameters $(\alpha_c, \eta_1, \eta_2)$ lead to different robust upper bounds but induce barycentric optima that lie on the same Wasserstein geodesic between $\mathbb{P}_c$ and $\mathbb{P}_{cf}$. In practice, we therefore bypass explicit bound tuning and instead select the barycentric weight $\lambda_c$ directly from the empirical geometry, yielding a CF-consistent representative of the robust equivalence class.

**Canonical barycentric classifier.** Given the surrogate pair $(\mathbb{Q}_0^*, \mathbb{Q}_1^*)$, we define the canonical barycentric classifier

$$\hat{m}(x) = \arg \min_{c \in \{0, 1\}} W_2(\delta_x, \mathbb{Q}_c^*).$$

**Corollary A.4** (Stability of the induced decision score). *Let $\mathbb{Q} = (\mathbb{Q}_0, \mathbb{Q}_1)$ and $\mathbb{Q}' = (\mathbb{Q}'_0, \mathbb{Q}'_1)$ be two pairs of class-conditional prototypes. Define the decision score*

$$\Delta_{\mathbb{Q}}(x) := W_2(\delta_x, \mathbb{Q}_1) - W_2(\delta_x, \mathbb{Q}_0),$$

*and the induced classifier*

$$\hat{m}_{\mathbb{Q}}(x) := \mathbb{I}[\Delta_{\mathbb{Q}}(x) \le 0].$$

*Then, for all $x \in \mathscr{X}$,*

$$\left| \Delta_{\mathbb{Q}}(x) - \Delta_{\mathbb{Q}'}(x) \right| \le W_2(\mathbb{Q}_0, \mathbb{Q}'_0) + W_2(\mathbb{Q}_1, \mathbb{Q}'_1).$$

*Consequently, for any L-Lipschitz loss $\phi : \mathbb{R} \to [0, 1]$,*

$$\left| \mathbb{E}_{x \sim \mathbb{P}_{\mathscr{X}}}[\phi(\Delta_{\mathbb{Q}}(x))] - \mathbb{E}_{x \sim \mathbb{P}_{\mathscr{X}}}[\phi(\Delta_{\mathbb{Q}'}(x))] \right| \le L(W_2(\mathbb{Q}_0, \mathbb{Q}'_0) + W_2(\mathbb{Q}_1, \mathbb{Q}'_1)).$$

**Proof.** By the triangle inequality of the Wasserstein distance,

$$\left| W_2(\delta_x, \mathbb{Q}_c) - W_2(\delta_x, \mathbb{Q}'_c) \right| \le W_2(\mathbb{Q}_c, \mathbb{Q}'_c) \quad \text{for } c \in \{0, 1\}.$$

Subtracting the two class scores yields the first inequality. The second inequality follows directly from the Lipschitz continuity of $\phi$ and Jensen's inequality. $\qquad\square$

**Remark.** If there exists a reference distribution $\mu_c \in \mathscr{C}_c$ such that $\mathscr{R}_c(\mathbb{Q}_c) \le \varepsilon_c$, then

$$W_2(\mathbb{Q}_c, \mu_c) \le \sqrt{\varepsilon_c}.$$

Hence, reconstruction error measured by $\mathscr{E}(\mathbb{Q}_0, \mathbb{Q}_1) = \mathscr{R}_0(\mathbb{Q}_0) + \mathscr{R}_1(\mathbb{Q}_1)$ controls the stability of the induced decision score up to $O(\sqrt{\varepsilon})$. This provides a geometric interpretation of why minimizing the barycentric objective promotes behavioral stability under CF-consistent uncertainty.

### A.4. Proof of Lemma 3.4

*Proof.* Fix $\lambda, \lambda' \in [0, 1]$ and define

$$g_\lambda(\mu) := W_2^2(\mu, Q_c^\star(\lambda)).$$

By definition of the supremum,

$$\left| R_c(Q_c^\star(\lambda)) - R_c(Q_c^\star(\lambda')) \right| = \left| \sup_{\mu \in C_c} g_\lambda(\mu) - \sup_{\mu \in C_c} g_{\lambda'}(\mu) \right|$$
$$\le \sup_{\mu \in C_c} \left| g_\lambda(\mu) - g_{\lambda'}(\mu) \right|.$$

For any fixed $\mu \in C_c$, let $a := W_2(\mu, Q_c^\star(\lambda))$ and $b := W_2(\mu, Q_c^\star(\lambda'))$. Using the identity $|a^2 - b^2| = |a - b|(a + b)$ together with the reverse triangle inequality for $W_2$, we obtain

$$\left| g_\lambda(\mu) - g_{\lambda'}(\mu) \right| \le (a + b) W_2(Q_c^\star(\lambda), Q_c^\star(\lambda')).$$

Next, since $C_c = \{\mu : W_2(\mu, P_c) \le \varepsilon_c, W_2(\mu, P_{cf}) \le \delta_c\}$, we have for any $\mu \in C_c$ and $\lambda \in [0, 1]$,

$$W_2(\mu, Q_c^\star(\lambda)) \le W_2(\mu, P_c) + W_2(P_c, Q_c^\star(\lambda)) \le \varepsilon_c + W_2(P_c, P_{cf}),$$

where the last inequality follows from the fact that $(Q_c^\star(\lambda))_{\lambda \in [0,1]}$ is a Wasserstein geodesic between $P_c$ and $P_{cf}$. An analogous bound holds for $W_2(\mu, Q_c^\star(\lambda'))$. Moreover, by the constant-speed property of the geodesic,

$$W_2(Q_c^\star(\lambda), Q_c^\star(\lambda')) = |\lambda - \lambda'| W_2(P_c, P_{cf}).$$

Combining these bounds yields

$$\left| g_\lambda(\mu) - g_{\lambda'}(\mu) \right| \le 2(\varepsilon_c + W_2(P_c, P_{cf})) W_2(P_c, P_{cf}) |\lambda - \lambda'|.$$

Taking the supremum over $\mu \in C_c$ concludes the proof. $\qquad\square$

---

**Algorithm 1** Optimization for discrete prototypes used in experiments

---

**Require:** Encoded point clouds $\mathscr{D}_0, \mathscr{D}_1, \mathscr{D}_{\mathrm{cf}} \subset \mathbb{R}^d$; support size $M$; iterations $T_{\max}$; learning rate $\eta$; random seed $s$;
    Sinkhorn parameters `blur`.
**Ensure:** Prototypes $\widehat{\mathbb{Q}}_0, \widehat{\mathbb{Q}}_1$.
  1: Set seed $s$.
  2: Define entropic OT surrogate $\widetilde{W}_2^2(\cdot, \cdot)$ using Sinkhorn with parameters `blur` and uniform weights for all discrete
    measures.
  3: **for** each class $c \in \{0, 1\}$ **do**
  4:    Derive $\lambda_c$ using $\widetilde{W}_2^2$ between $\mathscr{D}_{\mathrm{cf}}, \mathscr{D}_c$, and $\mathscr{D}_{1-c}$.
  5: **end for**
  6:
  7: **Initialization.** For each class $c \in \{0, 1\}$, randomly sample $M$ points without replacement from $\mathscr{D}_c$ to initialize the
    support locations $\mathbb{Q}_c \in \mathbb{R}^{M \times d}$; set support weights to uniform (fixed)
  8: Initialize Adam optimizer over support locations $\{\mathbb{Q}_0, \mathbb{Q}_1\}$ with learning rate $\eta$.
  9: **for** $t = 1$ to $T_{\max}$ **do**
10:    $L_0 \leftarrow (1 - \lambda_0)\, \widetilde{W}_2^2(\mathbb{Q}_0, \mathscr{D}_0) + \lambda_0\, \widetilde{W}_2^2(\mathbb{Q}_0, \mathscr{D}_{\mathrm{cf}})$.
11:    $L_1 \leftarrow (1 - \lambda_1)\, \widetilde{W}_2^2(\mathbb{Q}_1, \mathscr{D}_1) + \lambda_1\, \widetilde{W}_2^2(\mathbb{Q}_1, \mathscr{D}_{\mathrm{cf}})$.
12:    $\mathscr{L} \leftarrow L_0 + L_1$.
13:    Take an Adam step on the support locations of $\mathbb{Q}_0, \mathbb{Q}_1$ to minimize $\mathscr{L}$.
14: **end for**
15: Set $\widehat{\mathbb{Q}}_0 \leftarrow \mathbb{Q}_0$
16: Set $\widehat{\mathbb{Q}}_1 \leftarrow \mathbb{Q}_1$
17: **return** $\widehat{\mathbb{Q}}_0, \widehat{\mathbb{Q}}_1$

---

## A.5. Theoretical Guarantee of Convergence and Sinkhorn Divergence

Let $\mathbb{P}_0, \mathbb{P}_1, \mathbb{P}_{cf} \in \mathscr{P}_2(\mathscr{X})$ be probability measures with finite second moments supported on a compact metric space $\mathscr{X} \subseteq \mathbb{R}^d$ and $\lambda_c \in [0, 1]$ constants. Assume the optimization is carried out over the 2-Wasserstein space $W_2(\mathscr{X})$.

*Remark* A.5. Our loss function $\mathscr{L}$ defined as:

$$\mathscr{L}(\mathbb{Q}_0, \mathbb{Q}_1) = \sum_{c \in \{0,1\}} \left( (1 - \lambda_c) W_2^2(\mathbb{Q}_c, \mathbb{P}_c) + \lambda_c W_2^2(\mathbb{Q}_c, \mathbb{P}_{\mathrm{cf}}) \right),$$

admits a unique minimizer $(\mathbb{Q}_0^\star, \mathbb{Q}_1^\star) \in \mathscr{P}_2(\mathscr{X}) \times \mathscr{P}_2(\mathscr{X})$.

We can decompose the loss function $\mathscr{L}(\mathbb{Q}_0, \mathbb{Q}_1)$ in two independent parts

$$\mathscr{L}_{\mathbb{Q}_0} + \mathscr{L}_{\mathbb{Q}_1},$$

with

$$\mathscr{L}_{\mathbb{Q}_c} = \left( (1 - \lambda_c) W_2^2(\mathbb{Q}_c, \mathbb{P}_c) + \lambda_c W_2^2(\mathbb{Q}_c, \mathbb{P}_{\mathrm{cf}}) \right), \text{ for } c \in \{0, 1\},$$

which is essentially a version of a Wasserstein barycenter in Eq. (5) with $\mu = \mathbb{Q}_c$, $\lambda_i = \lambda_c, (1 - \lambda_c)$, $N = 2$, and $\mu_1$ and $\mu_2$ the probability measures with finite second moments $\mathbb{P}_c$ and $\mathbb{P}_{cf}$ respectively. It is also shown that the optimal solution for $p = 2$, is the geodesic curve provided by the McCann's interpolant (McCann, 1997; Agueh & Carlier, 2011). As a sum of two Wasserstein barycenters $\mathscr{L}(\mathbb{Q}_0, \mathbb{Q}_1)$ admits thus an optimal solution $(\mathbb{Q}_0^\star, \mathbb{Q}_1^\star)$.

**Approximation** Theoretical results in Sections 3.1- 3.3 are derived for the exact 2-Wasserstein distance $W_2$. In practice, we replace $W_2$ with the Sinkhorn divergence $S_\epsilon$ to obtain a smooth and computationally efficient objective using entropic regularization. This corresponds to optimizing a regularized (smoothed) approximation of the Wasserstein barycenter problem rather than the exact Wasserstein DRO objective. As $\epsilon \to 0$, the Sinkhorn divergence converges to $W_2$, and the solution approaches the true Wasserstein barycenter. For a regularization parameter $\epsilon$, the entropically regularized Wasserstein distance is defined as follows:

$$W_{\epsilon,2}^2(\mu, \nu) := \min_{\gamma \in \Pi(\mu,\nu)} \underbrace{\int \|y - x\|_2^2 d\gamma(x, y)}_{\text{transport cost}} + \underbrace{\epsilon H(\gamma | \mu \otimes \nu)}_{\text{relative entropy}},$$

Sinkhorn divergence is defined as:

$$S_\epsilon(\mu, \nu) := W_{\epsilon,2}^2(\mu, \nu) - \frac{1}{2}W_{\epsilon,2}^2(\mu, \mu) - \frac{1}{2}W_{\epsilon,2}^2(\nu, \nu).$$

The modified loss function is then

$$\mathscr{L}_\epsilon(\mathbb{Q}_0, \mathbb{Q}_1) = \sum_{c \in \{0,1\}} \left( (1 - \lambda_c)S_\epsilon(\mathbb{Q}_c, \mathbb{P}_c) + \lambda_c S_\epsilon(\mathbb{Q}_c, \mathbb{P}_{cf}) \right).$$

Approximation quality increases as $\epsilon \to 0$: $W_{\epsilon,2}^2(\mu, \nu) \to W_2^2(\mu, \nu)$, therefore the minimizers of $\mathscr{L}_\epsilon(\mathbb{Q}_0, \mathbb{Q}_1)$ converge to the minimizers of $\mathscr{L}(\mathbb{Q}_0, \mathbb{Q}_1)$.

Chizat et al. (2020) show that for $\epsilon \approx n^{-1/(d'+4)}$, they achieve with probability $1 - \theta$ that $|S_{\epsilon,n} - W_2^2| \lesssim n^{-2/(d'+4)} + n^{-1/2}\sqrt{\log(2/\theta)}$, with $n$ being the number of independent samples in $\mathbb{R}^d$ and $d'$ denoting $2\lfloor d/2 \rfloor$. When choosing $n \gtrsim \log(2/\theta)\varepsilon^{-(d'+4)/2}$, they achieve the desired $\varepsilon$-accuracy with $1 - \theta$. Thus they conclude a computational complexity of $O(n^2/(\epsilon\varepsilon))$.

## A.6. Sensitivity to Sinkhorn Regularization

The Sinkhorn algorithm introduces a regularization parameter $\epsilon$ that controls the trade-off between computational efficiency and approximation of the exact Wasserstein distance.

We ablate the blur parameter on Adult (query size $n = 50$) by sweeping the Sinkhorn blur ($\epsilon$) from 0.2 to 0.005. Fidelity remains stable across the range of values, while the reference-set mean margin $\mathbb{E}_x|W_2(x, Q_1^*) - W_2(x, Q_0^*)|$ decreases slightly from 82.66 to 81.08. This indicates that smaller blur leads to a modest reduction in margin, while the predicted labels remain largely unchanged, signaling stable decision behavior despite changes in regularization.

## A.7. Fairness

Let $\mathbb{P}$ denote the data-generating distribution over $(X, Y, S)$, where $S \in \{0, 1\}$ is the sensitive attribute, $X$ the random input variable, and $Y \in \{0, 1\}$ the class label as a random variable with realizations $y \in \{0, 1\}$.

Demographic parity (Dwork et al., 2012; Feldman et al., 2015) describes that the probability of an individual to be assigned to a class $y$ should not depend on the sensitive group $S$:

$$\textit{Demographic Parity} \quad P(\hat{Y} = y \mid S = 0) = P(\hat{Y} = y \mid S = 1).$$

Equalized Odds (Hardt et al., 2016) describes that the prediction accuracy should not depend on the sensitive group $S$:

$$\textit{Equalized Odds} \quad \begin{cases} P(\hat{Y} = 1 \mid S = 0, Y = 0) = P(\hat{Y} = 1 \mid S = 1, Y = 0) \\ P(\hat{Y} = 1 \mid S = 0, Y = 1) = P(\hat{Y} = 1 \mid S = 1, Y = 1). \end{cases}$$

We adapt *threshold-invariant demographic parity disparity* (TIDP) and *threshold-invariant equalized odds disparity* (TIEO) from Chen & Wu (2020). Specifically, we define *threshold-invariant demographic parity disparity* (TIDP) as the Wasserstein-1 distance between the score distributions conditioned on different values of $S$:

$$D_{\mathrm{DTIDP}} = W_1\left( \begin{array}{c} p_\# P(X \mid S = 0), \\ p_\# P(X \mid S = 1) \end{array} \right).$$

Analogously, *threshold-invariant equalized odds disparity* (TIEO) compares the score distributions conditioned on both $S$ and the label $Y \in \{0, 1\}$:

$$D_{\mathrm{DTIEO}}(y) = W_1\left( \begin{array}{c} p_\# P(X \mid S = 0, Y = y), \\ p_\# P(X \mid S = 1, Y = y) \end{array} \right).$$

These diagnostics capture disparities that persist across all decision thresholds, as they operate on entire score distributions. Moreover, they quantify how different sensitive groups are positioned relative to the barycentric decision geodesic.

## B. Experimental Setup

The following subsections deal with datasets, and implementation as well as experiment details.

### B.1. Description and Pre-processing of Real-World Benchmark Datasets

To evaluate our framework, we employ four well-known publicly available tabular datasets: **Adult Income**, **California Housing**, **COMPAS**, and **HELOC**. Below are their key characteristics:

- **Adult Income**: Derived from the 1994 U.S. Census, this dataset captures demographic and financial attributes such as education level, marital status, age, and annual earnings. The classification task involves predicting whether an individual's income exceeds \$50,000 (denoted as $y = 1$). The original dataset consists of 32,561 entries, with 24,720 labeled as $y = 0$ and 7,841 as $y = 1$. To balance the classes, we randomly selected 7,841 samples from $y = 0$, resulting in a final dataset of 15,682 entries. The dataset includes 6 numerical and 8 categorical features, with the latter converted to integer encodings. All features were normalized to $[0, 1]$.

- **California Housing Prices** (Housing): The dataset consists of data for houses in a district in California collected in the 1990 census. The dataset consists of 20,640 samples and 9 features. The target variable is the average house value, we transform it into a binary target variable by setting the median house value as a threshold.

- **COMPAS**: Developed to study racial bias in recidivism prediction algorithms, this dataset contains 6,172 entries with 20 numerical features. The target variable, `is_recid` divides the data into 3,182 ($y = 0$) and 2,990 ($y = 1$) samples. Feature values were normalized to $[0, 1]$.

- **Home Equity Line of Credit** (HELOC): This dataset records credit risk assessments for customers seeking home equity loans. It comprises 10,459 entries, each with 23 numerical features. The prediction target, `is_at_risk` identifies customers likely to default. The dataset is moderately imbalanced, with 5,000 samples for $y = 0$ and 5,459 for $y = 1$.

**Pre-processing.** We adopt a unified data loading and pre-processing pipeline across all datasets. For each dataset, we construct training, validation, and test splits using stratified sampling, holding out $20\%$ for testing and $20\%$ of the remaining data for validation. Target labels are standardized by stripping whitespace and removing trailing punctuation where applicable. Features containing missing indicators (e.g., "?", or dataset-specific sentinel values) are mapped to `NaN`, infinite values are removed, and rows with missing targets are discarded. Features are then automatically partitioned into continuous and categorical variables based on data type. Continuous features are median-imputed and scaled using a RobustScaler with the 5–95 percentile range, while categorical features are imputed with the most frequent value and one-hot encoded with unknown-category handling. All transformations are fit on the training split only and applied to validation and test data. We further store metadata such as output feature names, continuous and categorical index locations, one-hot category mappings, and scaler parameters, which are subsequently used both for model training and for decoding counterfactual examples back into raw feature space. For the California Housing dataset, we follow prior work and binarize the target at the median value to obtain a balanced binary prediction task.

*Importantly, we do not assume access to the target model's internal pre-processing pipeline.* While a unified pipeline is used within each experimental run to ensure internal consistency, the reconstruction model does not share or reuse the exact feature transformations employed by the target model. In this sense, pre-processing is treated as an integral part of the black-box system and is not exposed to the auditor.

We note that most existing model reconstruction and CF-based extraction methods implicitly assume that the attacker operates in the same feature representation as the target model, and therefore do not explicitly study the effect of pre-processing or representation misalignment. In practice, however, differences in normalization, encoding, imputation, or feature engineering can substantially alter the geometry of the input space and thereby impact reconstruction performance.

This distinction is particularly relevant when reproducing prior baselines: we observe that reconstruction fidelity can be noticeably lower than the values reported in earlier work, which we attribute in part to the absence of shared pre-processing assumptions. We view this setting as more reflective of realistic auditing scenarios, where the feature transformation pipeline is typically proprietary and inaccessible.

## B.2. Implementation Details

All experiments were implemented in Python 3.12 and conducted on a workstation equipped with an NVIDIA RTX 3090 GPU, with 124 GB of RAM running on Ubuntu 24.04. We used the following libraries for the implementation: `pandas`, `scikit-learn`, `geomloss`, `numpy`, `torch` and official code for the CF generation methods. Hyperparameters are set according to a grid search. Our code is included in the supplementary material.

## B.3. Experiment Details

The following sections present details regarding the experiment setup.

**Target classifiers.**    Across all datasets, we treat several standard tabular models as black-box target classifiers. Following the experimental protocol in the main paper, we train a logistic regression model (LR), a multilayer perceptron (MLP), and tree-based classifiers on the pre-processed feature representations obtained from our ColumnTransformer pipeline. The LR model is implemented as a single linear layer with a sigmoid output, optimized using Adam and a binary cross-entropy loss with logits, with early stopping on validation loss. The main MLP consists of two hidden layers with 20 and 10 units, respectively, with ReLU activations and a dropout rate of 0.1, followed by a single-unit output layer; it is trained with Adam, a learning rate of $10^{-3}$ and weight decay of $10^{-4}$, again with early stopping. For tree-based targets, we employ a decision tree (DT) with maximum depth 6 and minimum leaf size 20. All models are trained on the training split only and evaluated on the held-out test split using fidelity as the primary metric.

**Baselines.**    We compare our method against several representative reconstruction baselines that incorporate counterfactual explanations. Most of them are designed for interactive online access, however, since our setting is constrained to offline-only, we adapt the baselines and restrict them to offline data too. The first baseline, *SAMPLES*, follows Aïvodji et al. (2020) and simply augments the observed labeled samples with their CFs, treating CFs as ordinary labeled points when training standard classifiers such as LR, MLP and DT on the pre-processed features. The second baseline, *CCA* (Counterfactual Clamping Attack), implements the counterfactual clamping loss of Dissanayake & Dutta (2024): we train a small neural surrogate network with hidden layers of sizes $(20, 10, 5)$ and ReLU activations, followed by a sigmoid output, on tri-valued labels $y \in \{0, 0.5, 1\}$, where 0.5 denotes CFs. For regular samples ($y \in \{0, 1\}$) we use standard binary cross-entropy, while CFs are encouraged to have predictions above a threshold $k = 0.5$ via the clamping objective, which only penalizes CFs whose predicted probability does not exceed $k$. For completeness, we also report results for TRA (Khouna et al., 2025), which is competitive in our setting but restricted to tree-based models; we therefore instantiate TRA only for tree targets and include it as an additional baseline in the appendix.

**CF generation methods.**    We implement several CF generators in a unified framework. The *1-Nearest-Neighbor* (1NN) method searches over a pool of raw samples whose predicted label matches the desired outcome and selects the point with minimal mixed cost ($\ell_1$ distance on continuous features plus Hamming distance on categorical features). The differentiable *MCCF* variant optimizes a CF directly in encoded feature space using gradient-based updates on a scaled representation, with continuous dimensions updated freely and categorical groups projected to (approximate) one-hot vectors via straight-through Gumbel–Softmax; the objective trades off a prediction loss that pushes the output to the target class against an $\ell_1$-style proximity term. Our *DiCE* implementation follows a randomized search strategy in encoded space, sampling perturbations of continuous dimensions and randomly flipping categorical one-hot entries only for a specified subset of features, and returns the first candidate that achieves the desired prediction with minimal raw-space cost. For *C-CHVAE*, we train a tabular VAE in encoded space and then optimize in the latent space to find codes whose decoded samples both lie close to the original point (in encoded distance) and are classified as the target label; successful decoded candidates are mapped back to raw feature space via the inverse preprocessing pipeline. Finally, *ROAR* is implemented as a robust MCCF-style optimizer in encoded space replacing the standard prediction term with a robust probability under weight noise, aggregating predictions over multiple noisy parameter draws (either by averaging or worst-case aggregation), while using the same continuous and categorical distance regularizers; this yields CFs that remain valid under moderate target-model parameter shifts.

As shown in Fig. 2, CF generation methods exhibit substantial variability in the prediction scores assigned by the target model, with some overshooting the decision threshold. CCA assumes that CFs lie close to the decision boundary and leverages such near-boundary samples to shape the surrogate decision surface. When CFs instead yield prediction scores far above 0.5, this assumption is violated and reconstruction performance deteriorates. By contrast, RECAST neither interprets CFs as true samples (like *SAMPLES*) nor assumes proximity to the decision boundary. Instead, it incorporates CF

information through CF-consistent distributional geometry, which enables stable reconstruction even under heterogeneous CF generation behaviors.

**Wasserstein Barycenter Computation** We compute entropically regularized Wasserstein costs using the `SamplesLoss("sinkhorn", p=2)` implementation from GeomLoss (Feydy et al., 2019), which provides a differentiable Sinkhorn approximation of $W_2$ between empirical measures. To construct interpretable class representations, the experiment leverages 2-Wasserstein distances using the Sinkhorn algorithm implemented in the `geomloss` library. The objective is to learn two barycenters $\mathbb{Q}_0$ and $\mathbb{Q}_1$ that serve as prototypical representatives of each class in the transformed feature space. The loss function incorporates the following components: Wasserstein distance from class 0 samples to $\mathbb{Q}_0$ and $\mathbb{Q}_1$. Each barycenter is initialized with $M$ support points (default $M = 50$). Unless stated otherwise, all experiments use a fixed support size of $M = 50$. Optimization is performed using the Adam optimizer with a learning rate of 0.01 for 200 epochs. The formulation of the loss function follows Equation 3.2 from Section 3, with all Wasserstein distances replaced by their entropically regularized Sinkhorn surrogates.

**Hyperparameter Sensitivity.** We conduct an extensive ablation study to assess the sensitivity of RECAST with respect to its optimization and regularization parameters. Specifically, we perform a grid search over the learning rate $\eta \in \{10^{-3}, 5 \times 10^{-3}, 10^{-2}\}$, the number of training epochs $\{100, 200, 400\}$. In addition, since Wasserstein barycenters are computed via an entropically regularized Sinkhorn approximation, we evaluate the effect of the entropic regularization parameter `blur` $\in \{10^{-3}, 5 \times 10^{-3}, 10^{-2}, 5 \times 10^{-2}\}$.

Unless stated otherwise, we fix $\eta = 10^{-2}$, `blur` $= 5 \times 10^{-2}$, and train for 200 epochs in all experiments, which we found to provide stable numerical behavior in practice. Figure 8 illustrates the convergence behavior of the objective in Equation 3.2 on the Adult dataset under different query sizes $k$.

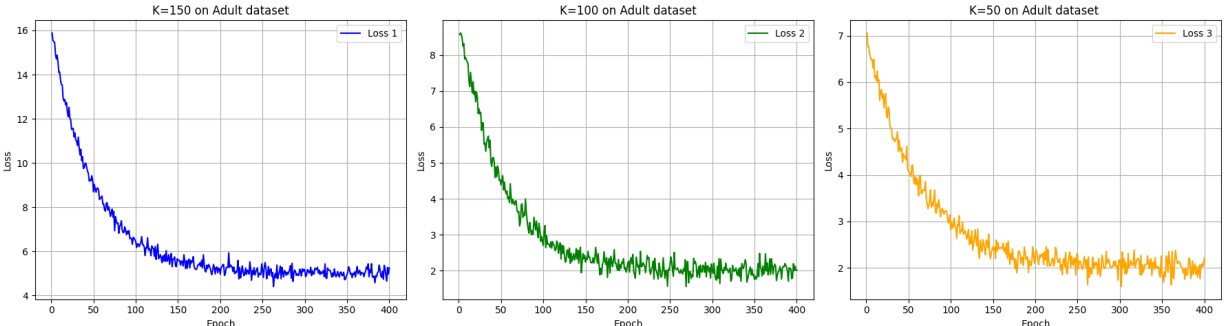

*Figure 8.* Loss convergence over epochs on Adult dataset.

**Repetition** All quantitative analyses were averaged across 10 independent repetitions to account for variability in sampling and model behavior.

# C. Additional Experiments

This section of the Appendix includes additional experiments, that could not be included in the main paper due to space limitations.

## C.1. Ablation Study

We conduct a ablation study to assess RECAST with respect to its design choices. First, we analyze the role of the class-conditional weighting $\lambda_c$ by selectively disabling each component. As shown in Table 4, removing either component degrades fidelity, while disabling both leads to substantial performance loss, highlighting their complementary roles. Second, we replace Wasserstein barycenters with other prototype constructions, such as the Euclidean mean, medoid-based prototypes and MMD. Here, CFs can only be included as labeled data. We emphasize that this difference is not a design choice but a structural limitation of the compared prototype representations. Euclidean mean and medoid prototypes operate on point estimates and do not admit a natural mechanism to incorporate CFs as soft, distributional constraints. These alternatives consistently underperform, particularly in low-query regimes, underscoring the importance of Wasserstein geometry for capturing CF-induced distributional structure. However, MMD can be plugged in our design easily with similar $\lambda_c$, with only Wasserstein distance changed to MMD. Finally, we evaluate the effect of counterfactual information by removing CFs from the reconstruction. As shown in Table 4, excluding CFs leads to a substantial drop in fidelity, supporting their role in shaping boundary-relevant geometry rather than pinpointing the boundary itself. Sensitivity to optimization hyperparameters (learning rate and training epochs) is reported in Appendix C.

*Table 4.* Ablation study of RECAST (Adult, MLP, query size = 100).

| Method Variant | CFs | Prototype | $\lambda_c$ | Loss ↓ | Fidelity (%) ↑ |
|---|---|---|---|---|---|
| RECAST (Full) | ✓ | Wasserstein | Ours | **2.18** | **90.8** |
| Fixed $\frac{\lambda_c}{1-\lambda_c}$ | ✓ | Wasserstein | $\frac{\lambda_c}{1-\lambda_c} = 0.5$ | 2.61 | 88.7 |
| Euclidean prototype | ✓ | Euclidean mean | – | 3.12 | 82.4 |
| Medoid prototype | ✓ | Medoid | – | 2.88 | 84.1 |
| Maximum Mean Discrepancy | ✓ | RBF-MMD | Similar to Ours | 2.65 | 87.3 |
| No CFs | ✗ | Wasserstein | – | 3.05 | 81.6 |

## C.2. Imbalanced Data

In our main experiments, we used the same value of query size for $\mathscr{D}_0$, $\mathscr{D}_1$, and $\mathscr{D}_{\mathrm{cf}}$. To evaluate the robustness of our approach under imbalanced data conditions, we conducted additional experiments using varying sample sizes for the counterfactual dataset. These experiments simulate more realistic, unbalanced settings in which the sizes of $\mathscr{D}_0$, $\mathscr{D}1$, and $\mathscr{D}\mathrm{cf}$ may differ, while the total number of natural samples satisfies $|\mathscr{D}_0| + |\mathscr{D}_1| = 200$. As shown in Table 5, our method maintains strong performance even under these imbalanced conditions.

*Table 5.* Fidelity under different class imbalance ratios.

| Dataset | 20% class 0 : 80% class 1 | | | 80% class 0 : 20% class 1 | | |
|---|---|---|---|---|---|---|
| | SAMPLES | CCA | RECAST (Ours) | SAMPLES | CCA | RECAST (Ours) |
| Adult In. | $0.751 \pm 0.017$ | $0.793 \pm 0.018$ | $\mathbf{0.848 \pm 0.030}$ | $0.702 \pm 0.016$ | $0.762 \pm 0.023$ | $\mathbf{0.819 \pm 0.027}$ |
| COMPAS | $0.385 \pm 0.007$ | $0.715 \pm 0.018$ | $\mathbf{0.795 \pm 0.020}$ | $0.352 \pm 0.011$ | $0.638 \pm 0.022$ | $\mathbf{0.738 \pm 0.025}$ |
| HELOC | $0.397 \pm 0.014$ | $0.604 \pm 0.052$ | $\mathbf{0.646 \pm 0.071}$ | $0.365 \pm 0.016$ | $0.561 \pm 0.045$ | $\mathbf{0.592 \pm 0.059}$ |
| Housing | $0.429 \pm 0.008$ | $\mathbf{0.666 \pm 0.065}$ | $0.662 \pm 0.112$ | $0.391 \pm 0.010$ | $\mathbf{0.624 \pm 0.057}$ | $0.615 \pm 0.088$ |

## C.3. Experiments including Accuracy

Table 6 reports target model accuracy alongside surrogate accuracy and fidelity for RECAST, SAMPLES, and CCA on the Adult and COMPAS datasets at query sizes $n = 100$ and $n = 150$, using logistic regression as the target model and 1-nearest-neighbor as the CF method. We report both metrics because they capture different objectives: accuracy measures predictive performance against ground-truth labels, whereas fidelity measures agreement with the target model's predictions, including its errors. Consequently, the two metrics need not correlate, a surrogate can achieve high fidelity while having lower task accuracy if the target model itself makes systematic errors. RECAST achieves the highest fidelity across both

*Table 6.* Target model accuracy, surrogate accuracy, and fidelity at query sizes $n = 100$ and $n = 150$ on Adult and COMPAS. Accuracy and fidelity need not correlate: fidelity measures agreement with the target model's predictions (including its errors), not ground-truth label accuracy. Best surrogate results per dataset and query size in **bold**.

| Dataset | Method | Target Acc. | $n = 100$ | | $n = 150$ | |
|---|---|---|---|---|---|---|
| | | | Acc. | Fid. | Acc. | Fid. |
| Adult | Samples | 0.8463 | 0.7463 | 0.7800 | 0.7488 | 0.7679 |
| | CCA | 0.8463 | 0.5598 | 0.5097 | 0.5318 | 0.4797 |
| | RECAST | 0.8463 | **0.7741** | **0.8400** | **0.7700** | **0.8600** |
| COMPAS | Samples | 0.6696 | **0.5983** | 0.7004 | 0.5424 | 0.7518 |
| | CCA | 0.6696 | 0.5405 | 0.5441 | 0.5376 | 0.5398 |
| | RECAST | 0.6696 | 0.5580 | **0.7560** | **0.5770** | **0.7860** |

*Table 7.* Accuracy and fidelity on Folktables (ACSIncome) under cross-domain distribution shift (train: CA 2018, evaluate: MI 2014) at query sizes 100 and 150.

| Reconstruction | Query size = 100 | | Query size = 150 | |
|---|---|---|---|---|
| | Acc. | Fid. | Acc. | Fid. |
| SAMPLES | **0.7356** | 0.8641 | **0.7499** | 0.8596 |
| CCA | 0.7302 | 0.6362 | 0.7251 | 0.6251 |
| RECAST | 0.6473 | **0.8661** | 0.7254 | **0.8886** |

datasets and query sizes, while remaining competitive on accuracy, demonstrating that behavioral agreement with the target is not achieved at the expense of predictive performance.

## C.4. Additional Experiments on Distribution shifts

To investigate robustness under realistic distribution shift, we conduct an additional experiments using Folkstables (Ding et al., 2021). We train a logistic-regression target model on ACSIncome data from California (CA), 2018 (acc=0.7860), and evaluate reconstruction under shift to Michigan (MI), 2014, a setting that combines both geographical and temporal variation. We generate CFs using 1-nearest-neighbour and evaluate the performance (fidelity and accuracy) on query budget (100, 150).

Results are shown in Table 7. RECAST consistently achieves higher fidelity across query budgets (100 and 150), demonstrating robust recovery under distribution shift. SAMPLES leads on accuracy but trails substantially on fidelity, consistent with the pattern observed in Section C.3, accuracy and fidelity capture different objectives, and fidelity is the primary measure of reconstruction quality. These results demonstrate that RECAST maintains strong behavioral consistency even under substantial distribution shifts.

## C.5. Crossover Between RECAST and No-CF Reconstruction

We investigate when CF supervision becomes less critical for model reconstruction under increasing data availability. Specifically, we compare RECAST with a no-CF baseline trained solely on natural samples, as the number of available samples grows.

We fix a target model trained on the Adult dataset and evaluate reconstruction fidelity, defined as the fraction of test instances on which the surrogate and the target model produce identical predictions. For the no-CF baseline, we use a decision tree surrogate, which exhibits stable behavior in low-sample regimes while allowing increased expressivity as more data become available.

Figure 9 illustrates a crossover pattern. In low-data regimes, CF-based reconstruction achieves substantially higher fidelity, highlighting its strong sample-efficiency advantage. As the number of natural samples increases, the performance gap narrows and reverses around $n \approx 1000$ samples per class, where the no-CF surrogate attains comparable or higher fidelity.

This pattern reflects a trade-off between data efficiency and surrogate expressivity in audit-oriented reconstruction settings. Counterfactual supervision provides informative local constraints when observational data are scarce, but yields diminishing benefits once sufficient coverage of the input space is available.

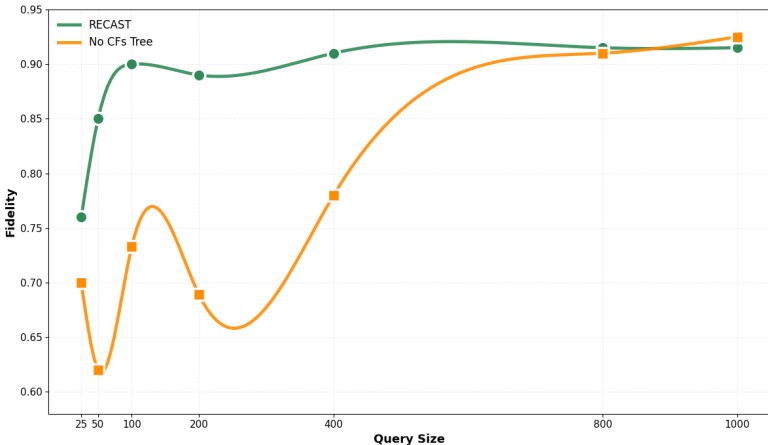

*Figure 9.* Crossover between CF-based and no-CF reconstruction on the Adult dataset. CF-based reconstruction achieves higher fidelity in low-data regimes, while the no-CF decision tree surrogate improves with increased natural samples and surpasses the CF-based method around $n \approx 1000$ per class.

### C.6. Data Modalities / High-Dimensional Data

Our framework is generally applicable to any data modality that can be embedded as vectors, e.g., pixel representations of visual data. Thus, we can apply RECAST to high-dimensional datasets after transforming the input data to latent space.

To show this, we perform an additional proof-of-concept on the well-known MNIST dataset (Deng, 2012). We load the dataset from the sklearn library, normalize the pixel values to be in-between $[0, 1]$ and reduce the classification problem to binary (e.g., classification between 0 and 1). For the target classifier, we employ 66% to train a standard sklearn MLP-classifier and generate CFs as the nearest neighbor belonging to the other class. RECAST is then trained on a subset of the training set in pixel-space ($d = 784$). Across different binary tasks and sample sizes (around 100 - 500), RECAST consistently achieves a fidelity around 0.7 or higher, see Figure 10.

### C.7. Language Case Study: Reconstruction under One-Sided Textual CFs

We present an exploratory case study on reconstruction for text-based classifiers under limited access. The target model is a binary classifier $m : \mathbb{R}^d \to \{0, 1\}$ trained on fixed text embeddings. Texts $x \in \mathscr{X}$ are mapped to embeddings $\psi(x) \in \mathbb{R}^d$ using a frozen LLM-based encoder. During reconstruction, the target model is accessed only through discrete label queries $m(\psi(x))$.

**Representation space.** Since optimal transport is not defined over raw text, all reconstruction is performed in the embedding space induced by $\psi$. The embedding model is fixed and shared across all methods.

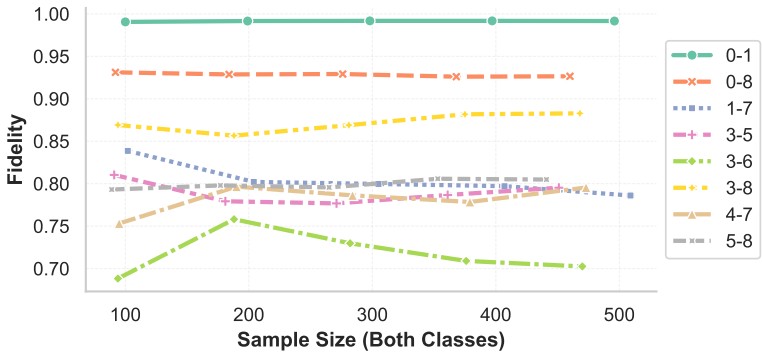

*Figure 10.* Fidelity results for various sample sizes and binary classification problems within the MNIST dataset.

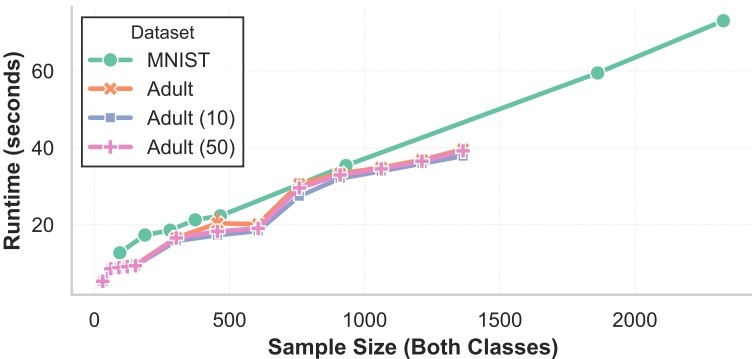

*Figure 11.* Runtime for RECAST across different datasets and configurations. Each RECAST model is trained for 500 epochs.

**Task and target model.**    We consider a toxicity classification task ([Jigsaw, 2018](#)). A binary classifier $m$ is trained on embeddings $\psi(x)$ obtained from a standard toxicity dataset and is treated as a black-box model during reconstruction.

**One-sided textual CFs.**    To emulate one-sided recourse, CFs are generated only for toxic instances with $m(\psi(x)) = 0$. For each such input $x$, a single textual CF $x^{\text{cf}}$ is produced using an LLM-based rewriting prompt that aims to minimally modify the text toward a non-toxic outcome. The resulting text is embedded via $\psi$, and the CF is retained only if $m(\psi(x^{\text{cf}})) = 1$. This yields three offline datasets: $\mathscr{D}_0 = \{x : m(\psi(x)) = 0\}$, $\mathscr{D}_1 = \{x : m(\psi(x)) = 1\}$, and $\mathscr{D}_{\text{cf}} = \{x^{\text{cf}} : x \in \mathscr{D}_0\}$.

**Reconstruction .**    We apply RECAST in the embedding space to compute class prototypes $\mathbb{Q}_0$ and $\mathbb{Q}_1$ using the barycentric objective.

**Evaluation.**    We evaluate query budgets of 100, 150, and 200 samples per class. Reconstruction fidelity is consistently lower than in tabular settings, typically saturating around $80\%$. We attribute this to strong representation compression: text inputs are mapped to fixed embeddings and further reduced to a 50-dimensional space.

## C.8. Runtime

To investigate runtime, we vary the number of samples used to train RECAST. The runtime experiments are performed on a MacBook Pro with 24 GB RAM and an Apple M4 Pro. We first split the full dataset in 33% test and 67% training data. Then we take different fractions, of the training set to train RECAST. We include MNIST ($d = 784$) and three versions of adult ($d = 14$ before pre-processing, $d = 104$ with one-hot-encoding) with varying feature sets ($d = 104, d = 50, d = 10$) to account for different sample dimensionalities. Each RECAST instance is trained for 500 epochs. Note that implementation improvements like early stopping could improve these runtimes, however we want to focus on a fair comparison and trained each instance the same amount of epochs. To mitigate influence of background tasks, we performed 5 runs per configuration. While the complexity of our approach is high in theory, we observe, see Figure 11, that the runtime in practice grows near linear with the number of samples. We additionally see that the number of features, compare adult with 104, 50, and 10 features, has even less impact on the runtime.

