# OpenReview forum: "RECAST: Model Reconstruction via Counterfactual-Aware Wasserstein Geometry under Limited Data"
_ICML.cc/2026/Conference — ICML 2026 regular_

### Official Review · Reviewer_ysDJ · 2026-03-10

**Soundness:** 4
**Presentation:** 3
**Significance:** 3
**Originality:** 3
**Overall Recommendation:** 3
**Confidence:** 4

**Summary:**

This paper proposed RECAST, an algorithm to reconstruct black-box binary classifiers in highly-constrained auditing conditions, including offline data and access to only one-side counterfactuals (CFs). RECAST uses Wasserstein barycenters (L2) to create class prototypes, treating CFs as soft distributional constraints rather than hard boundary points, which mitigates the boundary shift and overfitting issues seen in prior works.

**Compliance With Llm Reviewing Policy:**

Affirmed.

**Final Justification:**

I appreciate the authors' follow-up. I will increase my Soundness and Significance scores, as I accept the low-sample defense and the rigorous Wasserstein formulation. However, I am maintaining my overall score due to structural limitations that severely restrict the method's practical utility. First, the assumption that preprocessing "preserves relative geometric structure" is highly brittle; real-world proprietary pipelines routinely use non-linear transformations (e.g., log-transforms, binning) that distort relative $L_2$ distances. For example, a log-transform on income heavily compresses the relative distance between high earners compared to low earners, which would directly misalign the surrogate's barycenter. Second, as acknowledged, extending this beyond strictly binary classification to multi-class tasks requires a fundamentally novel formulation.

**Key Questions For Authors:**

1. Figure 2 shows that CFs often overshoot the decision threshold. If a CF generator systematically overshoots by a massive margin, wouldn't the barycenter be dragged too far into the opposing class's region and distort the surrogate's decision boundary? How to fix or calibrate it?
2. Wasserstein distances are exceptionally sensitive to feature scaling and metric spaces. If an auditor's external pre-processing drastically differs from the target's proprietary pipeline, the geometric assumptions of the barycenters could completely collapse? How to fix or detect it?
3. Real-world auditing frequently involves multi-class decisions or continuous risk scoring (regression). How to extend the current framework?

**Limitations:**

1. As the method is built upon the W2 distance, it means that the method is highly sensitive to the underlying ground metric (e.g., Euclidean distance). This requires strict, high-quality feature normalization, which may not align with the black box's internal scaling
2. The framework is explicitly limited to binary classification. The authors should extend the Wasserstein barycentric prototypes to multi-class scenarios where CFs could point toward multiple different target classes
3. The paper defines its query budget by the final available dataset size. However, generating a single valid CF against a true black-box API often requires dozens or hundreds of hidden queries [1]. Does it violate the low-sample claims? What if we consider the cost of acquiring the CFs?
4. Optimal transport calculations scale poorly. How would RECAST perform computationally on a moderate dataset (e.g., 10k+ samples)? Current experiments are mostly on small sample size and number of features.

[1] Singla, S., Eslami, M., Pollack, B., Wallace, S. and Batmanghelich, K., 2023. Explaining the black-box smoothly—a counterfactual approach. Medical image analysis, 84, p.102721.

**Strengths And Weaknesses:**

The authors did a great job on providing a strong theoretical results on the non-identifiability of decision boundaries with one-sided CFs. Formulating the reconstruction as a minimization of worst-case distributional risk using Wasserstein Robust Optimization provides a rigorous mathematical foundation.

For Weaknesses, please refer to Key Questions and Limitations.

---

> ### Author Rebuttal · Authors · 2026-03-31
>
> Dear Reviewer,
>
> Thank you very much for your time and valuable feedback.
> Before addressing each point individually, we briefly restate the problem setup to provide a shared frame of reference: our goal is to reconstruct the decision behavior of a black-box target model using only a limited offline query budget of samples and counterfactuals, without access to pre-processing, the CF generator, or any model internals. The central metric fidelity compares predictions between target and surrogate, thereby implicitly capturing the effects of differing preprocessing as well.
>
>
> - **Q1 (overshooting CFs and distortion):** The short answer is ‘No’. This is a valid concern with established classifier reconstruction methods and precisely the motivation for our design. Since CFs may overshoot the decision boundary, we do not treat them as reliable boundary samples or hard opposite-class evidence. Instead, their influence is mediated through the barycentric mixing weight $\lambda_c$, so the prototype $Q_c$ is determined by a compromise between $P_c$ and $P_{cf}$ rather than being directly pulled toward $P_{cf}$. As a result, when CFs are geometrically inconsistent with class $c$, their effect is reduced, and the reconstruction remains anchored to the empirical class distribution. In the extreme case where the CF distribution aligns with the opposite class (e.g., $P_{cf} \approx P_1$), the formulation reduces to recovering the empirical class distributions ($Q_0^* = P_0$, $Q_1^* = P_1$), corresponding to a standard classification.
>
>
> - **Q2 + L1 (sensitivity to pre-processing):**  Like all distance-based approaches, we agree that our approach based on Wasserstein distances also depends on the underlying feature space. However, our formulation does not rely on recovering an absolute geometry. Instead, the robust optimization problem is absorbed into the barycentric objective via the parameter $\lambda_c$, which encodes the relative geometric relationship between empirical and counterfactual distributions. As a result, the reconstruction depends on the relative positioning of these distributions rather than their absolute scale. Therefore, as long as preprocessing preserves the relative geometric structure between these distributions, the method remains stable. This makes our approach less sensitive to preprocessing compared to methods that rely directly on individual CF samples as boundary points.
> This is also supported empirically in Appendix B, where our experiments do not assume access to the preprocessing of the target model.
>
>
> - **Q3 + L2 (extension to regression and multi-class):** We agree that multi-class and regression settings are practically relevant, particularly in real-world auditing scenarios. However, extending the framework is non-trivial: in the binary case, each CF implicity provides signal for both classes, whereas in multi-class settings, a CF carries a directional relationship, that indicates a specific class-to-class transition, which requires to fundamentally rethink and redesign concepts for mixing weights and barycentric prototypes. Simply replicating the binary structure for each pair of classes would not preserve the theoretical properties of RECAST. As noted in our conclusion, we consider extending to multi-class and regression an interesting direction for future work.
>
>
> - **L3 (hidden queries for CF generation):** Thank you for raising this question. We would like to clarify that CF generation occurs on the provider's side; that is to say, the provider generates CFs using their own access to their model. These hidden queries are therefore not included in the outsider’s query budget. Consequently, this does not violate the low-sample regime, since the outsider's budget remains unaffected by CF generation and is only used to create the offline dataset. The underlying assumption is that Machine Learning-as-a-Service providers are obliged to provide a 'reject-to-accept' CF for each rejected sample. That said, we as outsiders do not have additional knowledge as to how the generator works, how many hidden queries it requires, or the cost of acquiring the CFs.
>
>
> - **L4 (scalability larger datasets):** This is a natural question given the known computational cost of optimal transport. We conducted runtime experiments with dimensionalities of up to 784 in Appendix C.4. These experiments suggest that the dimensionality of the datasets has empirically less dominant impact on runtime than sample size, whereas the sample size is the driving factor. The maximum sample size investigated here is approximately 2,400. Importantly, however, our approach is specifically designed for a low data regime, and our experiments demonstrate that RECAST achieves high fidelity well before all available samples in the included datasets are used. Scaling to 10k+ samples would therefore not test a meaningful scenario but rather shift the problem setting toward a setting where different design assumptions and baselines apply.

---

> > ### Author Rebuttal · Reviewer_ysDJ · 2026-04-03
> >
> > I thank the authors for their detailed rebuttal, particularly for clarifying the provider-side assumption regarding counterfactual generation costs (L3) and explaining how the mixing weight $\lambda_c$ relies on relative rather than absolute geometry to help mitigate preprocessing discrepancies (Q2, L1). However, I have decided to maintain my score of 3 because the practical applicability of the proposed framework remains a significant concern. By explicitly restricting the algorithm to low-data regimes instead of proving its feasibility on moderate datasets (L4), and by acknowledging that extensions to multi-class or continuous tasks are non-trivial and left to future work (Q3, L2), the method's real-world auditing utility is inherently limited. While the paper's mathematical and theoretical foundations are undeniably strong, its current scope is simply too narrow to provide the broader methodological impact required for a strong acceptance.

---

> > > ### Author Response · Authors · 2026-04-06
> > >
> > > We sincerely thank the reviewer for the continued engagement and acknowledging the strength of the mathematical and theoretical foundations. While we respectfully hold a different view on the scope of the paper's contribution, we appreciate the opportunity to offer a final clarification of our perspective.
> > >
> > > We would also like to clarify that the limited-query, low-data regime (L4) is not a limitation of our method, but a defining characteristic of the reconstruction and auditing setting. In many practical scenarios involving sensitive decision systems, such as credit approval, hiring, or admissions, access to data is inherently restricted due to privacy and regulatory constraints, and decisions are often binary in nature. As a result, the goal is precisely to achieve high-fidelity reconstruction under such constraints. Our method is explicitly designed for this regime, and its effectiveness in low-query settings should therefore be viewed as addressing the core challenge of the problem rather than reflecting limited applicability.
> > >
> > > Our Appendix C.4 already includes a proof-of-concept experiment on the multiclass  (Q3, L2) MNIST dataset, where we evaluate RECAST across multiple binary class-pair tasks (e.g., 0–1, 3–8, 5–8). Each of these tasks can be interpreted as a directed class-pair reconstruction problem within a multiclass setting, preserving the one-sided CF semantics. We did not emphasize this experiment as a full multiclass evaluation, since it does not constitute a principled multiclass formulation, but rather a collection of pairwise probes. In particular, naively aggregating counterfactual signals across multiple classes may obscure the directional structure inherent in counterfactual transitions. A more natural extension, which we view as a promising direction for future work, is to explicitly model these directional relationships via a graph-based formulation over classes. In this view, the current binary formulation can be seen as a fundamental building block, capturing the core geometric mechanism that would underlie such extensions.
> > >
> > > We appreciate the reviewer's thoughtful feedback, which has helped us both improve the clarity of our presentation and further sharpen the positioning and future directions of this work.

---

### Official Review · Reviewer_Banq · 2026-03-13

**Soundness:** 3
**Presentation:** 3
**Significance:** 3
**Originality:** 3
**Overall Recommendation:** 5
**Confidence:** 2

**Summary:**

The paper studies black-box model reconstruction under a restrictive but practical setting: offline access to labeled inputs together with only one-sided counterfactual explanations. It argues that in this regime, exact decision-boundary recovery is fundamentally non-identifiable, so the goal should be behavioral consistency with the data. The paper proposes RECAST, which builds class prototypes as Wasserstein barycenters between empirical class-conditional distributions and the counterfactual distribution, and then classifies points by distance to these barycentric prototypes. The paper also adapts threshold-invariant fairness metrics (TIDP and TIEO) to this geometric framework for auditing purposes. Experiments on four binary tabular datasets show improved fidelity over adapted baselines, especially in low-query and noisy settings.

**Compliance With Llm Reviewing Policy:**

Affirmed.

**Final Justification:**

Thanks for the author response. I remain uncertain/skeptical that there are not clear analogs in classical prediction with reweighting (for example under data selection bias) and do not think that comparing against the simplest cases (as the authors did in their rebuttal by comparing to iid prediction or SVMs) is fair to the rich literature that exists. However, I do think the paper is strong and will raise my score.

**Key Questions For Authors:**

In the list above

**Limitations:**

Limitations discussion looks good

**Strengths And Weaknesses:**

Strengths
1. The problem setting is meaningful and well motivated. The paper clearly explains why one-sided CF access and offline reconstruction are realistic in auditing settings, and why direct boundary recovery is hard under these constraints.
2. The paper has a good conceptual contribution. Reconstructing class-level behavior via Wasserstein geometry is interesting.
3. The baseline settings are reasonable, and the results show the performance of RECAST.

Weaknesses:
1. The introduction is a bit meandering. The authors didn't spell out CFs in the intro (only in the abstract), and they motivate it via fairness which seems like an application of the broader method but not central to the overall goal of reconstruction.
2. I had difficulty understanding Figure 1. In this figure, what indicates that there should be overconfidence? (The authors might want to clarify that the CFs are for the blue points, if I understand correctly)
3. The motivation of the work and how it is situated relative to the literature feels incomplete. The paper focuses on the much more recent literature of MAEs, but it seems like there is a long line of statistical learning literature that would be relevant. In the end, the goal is basically the same as machine learning: assuming that the real model/true decision boundary is “nature”, learn an ML model that is close to nature (the model that produces the data and CFs). The only difference between this and machine learning is semantic: that the “world” from which the sample are drawn are actually the original model. It seems that the analysis should be therefore more closely tied to the long literature of learning from limited samples (very early ML), offline learning, etc. There is the CF component which is to say that the samples are not i.i.d. but that should actually make learning even easier because it gives a strong indication of where the boundary is (which the authors do say “CFs provide richer information than label queries”). Maybe it’s also related to SVMs?Please correct me if I am missing something.
4. Following the results, I had trouble understanding the novelty. This could be that the contribution is very novel but the relation to the literature is not clear. Would the authors be able to clarify this (with mathematical precision, ideally with some indication of the novelty of the formal results)? For example, the theorem seems like it might standard from DRO? The observation of the ratio is nice and clean; it seems like a propensity scoring approach, where the reweighting accounts for bias of CFs?
5. Some of the concepts would benefit from being more clearly defined. For instance, “in this regime, λc is not uniquely identifiable” is a throwaway statement on page 4 that does not seem supported or formalized. Relate to point 3 above, most ML problems are not identifiable with limited data as well, so this seems like renaming of a well accepted concept?

I'm open to raising my score but believe the paper needs to be more well contextualized and written a bit more clearly. Without it, the paper is compelling but potentially not at ICML standard.

---

> ### Author Rebuttal · Authors · 2026-03-31
>
> Dear Reviewer,
>
>
> Thank you very much for your time and insightful feedback. In the following we will clarify and address your concerns. The concern stems from interpreting our setting as standard supervised learning. We appreciate your comments to improve our presentation, and will carefully revise the final version to address them.
>
> - **W1 (introduction):** Thank you for pointing this out! We will amend the definition accordingly and focus the description on the use of CFs for reconstruction in our work.
>
> - **W2 (Figure 1 - overconfidence):** As identified correctly, Figure 1 illustrates CFs (X) for class 0, representing the minimal changes needed to flip the target’s prediction from class 0 to class 1. These CFs are thus located close to the target model's decision boundary (dashed line). If these were naively used as samples for class 1, the learned boundary would shift toward class 0 (solid line), causing original class 0 samples to be incorrectly classified as class 1. We refer to this as 'overconfidence' since the surrogate becomes overly certain about class 1 in regions where the target model is not. We will amend the figure caption to clarify that the CFs are for class 0 samples.
>
>
> - **W3 (relation to ml /SVMs):** The similarity to classical small-sample learning is only superficial. The distinction is not semantic. Classical learning assumes a well-defined and identifiable target function, approximated via empirical risk minimization on i.i.d. samples with regularization or margin-based methods. In identifiable settings, margin-based methods such as SVM rely on samples near the decision boundary as informative signals. In contrast, in our setting CFs are not reliable boundary samples and may overshoot or be biased (Fig. 2), so such margin-based interpretations do not apply, leading to a fundamentally different regime.
> CFs do provide richer information than label queries, in the sense that they encode directional information as relative constraints in our setting. Our approach therefore does not rely on precise placement of CFs near the boundary, but instead incorporates their geometric information at the distribution level via Wasserstein-based surrogates.
> Empirically (see response to Reviewer G6uh), accuracy and fidelity are not necessarily correlated, further highlighting that reconstruction is a distinct objective from classical prediction.
>
>
> - **W4 (novelty):** We appreciate this question and agree that the relationship to existing DRO literature deserves clearer exposition. A more detailed discussion was originally included but omitted due to space constraints, and will be reinstated in the revised version. The core novelty lies in deriving a tractable surrogate objective from a CF-consistent uncertainty set whose minimizer yields a Wasserstein barycenter structure, which is not a direct application of standard DRO. Specifically, standard WRO formulations optimize over one Wasserstein ball calibrated with a radius around an empirical distribution. In our setting, the ambiguity radii cannot be reliably calibrated from limited data, instead we work with CF-consistent uncertainty sets that constrain the geometry. Our upper bound absorbs the radii into the class-specific mixing weights $\lambda_c$ and its minimizer recovers Wasserstein barycenters as distributional prototypes. The mixing weight is therefore not a propensity-style reweighting, but a geometry-induced surrogate that replaces infeasible radius calibration in this setting.
>
>
> - **W5 (definitions / throwaway statement):** We agree that the term “identifiable” was used imprecisely and might cause confusion. We mean it cannot be reliably calibrated from limited one-sided CF data, whereas in classical WRO the radii would be statistically calibrated and identifiable with sufficient samples. We will make this more explicit in the revised version of the paper.

---

> > ### Author Rebuttal · Reviewer_Banq · 2026-04-04
> >
> > Thanks for the author response. I remain uncertain/skeptical that there are not clear analogs in classical prediction with reweighting (for example under data selection bias) and do not think that comparing against the simplest cases (as the authors did in their rebuttal by comparing to iid prediction or SVMs) is fair to the rich literature that exists. However, I do think the paper is strong and will raise my score.

---

> > > ### Author Response · Authors · 2026-04-06
> > >
> > > We thank the reviewer for the constructive feedback and acknowledging the strengths of our approach. We appreciate the point regarding connections to classical reweighting under data selection bias, and we agree that a more thorough discussion of these connections would strengthen the paper. We will expand the discussion in the related work to position RECAST more carefully in relation to this literature in the revised version.

---

### Official Review · Reviewer_3Hmb · 2026-03-13

**Soundness:** 2
**Presentation:** 3
**Significance:** 3
**Originality:** 3
**Overall Recommendation:** 5
**Confidence:** 3

**Summary:**

This paper proposes RECAST, a model reconstruction method using counterfactual explanations (CEs).
When only one-sided CFs are available, traditional CE-based reconstruction methods suffer from biased decision boundary shifts and overfitting.
To address this, the proposed method avoids constructing a direct boundary and considers a behavior-centric approach that constructs class-wise predictive distributions.
Specifically, the authors define an ambiguity set from observed data and CF examples. They reformulate the problem, minimizing the worst-case risk under this ambiguity set as a problem for computing the 2-Wasserstein barycenter. For new inputs, the model predicts labels by comparing Wasserstein distances to the constructed class prototypes. Numerical experiments demonstrate that the method effectively reconstructs predictive models even under low-query and noisy environments.

**Compliance With Llm Reviewing Policy:**

Affirmed.

**Final Justification:**

Since the authors carefully addressed my concerns, I have updated my evaluation from Weak Accept to Accept.

**Key Questions For Authors:**

1. How can Problem (1) be interpreted from a robust optimization perspective, given that it is independent of the ambiguity set’s radii? Specifically, how do the authors justify the interpretation of this model in scenarios where the ambiguity set can be empty?
2. Could the authors discuss the relationship between the Sinkhorn algorithm’s regularization term (blur parameter) and the reconstructed model’s predictive performance? It would be helpful to see how sensitive the performance is when the regularization parameter is adjusted toward the exact Wasserstein distance.

**Limitations:**

yes

**Strengths And Weaknesses:**

Strengths:

- The motivation is clear, effectively introducing the proposed approach based on the Wasserstein distance.
- The paper proposes a reconstruction method under the strict limitation of one-sided CE availability.
- The stability analysis using Lipschitz continuity to evaluate the impact of the weight parameter ($\lambda$) is interesting.
- The numerical experiments are comprehensive, and their detailed settings are clearly written.
- Achieving high fidelity with a limited number of queries and maintaining stable fidelity against data distribution shifts are notable.

Weaknesses:

- Problem (1) is independent of the radius parameters ($\varepsilon_c$, $\delta_c$) for the ambiguity set. Consequently, the size of the ambiguity set does not affect the constructed prediction model, which makes the logical connection between the original robust optimization model and Problem (1) unclear.
- It is not intuitive for me that the Problem (1) remains valid even when we set $\varepsilon_c = \delta_c = 0$, where the ambiguity set can be empty. Given this weak connection to the original robust optimization formulation, a more rigorous discussion is required to explain why the proposed method exhibits robustness to noisy data than existing baselines.
- The paper lacks a concrete discussion of the mechanisms underlying why the prototype-based structure suppresses overfitting and achieves high fidelity when the number of queries is small.

---

> ### Author Rebuttal · Authors · 2026-03-31
>
> Dear Reviewer,
>
> Thank you very much for your time and effort to provide us with valuable feedback to improve our paper!
>
> - **W1 (logical connection):** We agree that, in classical WRO, the ambiguity radii should influence the solution. This is also our view. Our point is not that the radii are irrelevant, but that under one-sided CF access and limited data, they cannot be reliably calibrated from samples. In our derivation (details in App A.3), their relative effect is absorbed into the mixing weight $\lambda_c$, and selecting an optimal $\lambda_c$ would indeed correspond to selecting optimal radii. Since such calibration is statistically infeasible here, we instead choose a geometrically motivated $\lambda_c$​, which yields a stable and suboptimal (line 249 to 251) solution.
>
> - **W2+Q1 (robustness to noise / empty ambiguity set):** Unlike standard Wasserstein DRO, our objective is not to directly solve the constrained robust problem, but to optimize a tractable worst-case upper bound over the ambiguity set (page 4 left side). This upper bound is valid under the assumption that the radii are chosen appropriately so that the feasible sets are non-empty (line 183). The case of an empty ambiguity set thus falls outside the scope of our formulation and does not invalidate the formulation rather than representing a failure case. Importantly, the formulation degrades gracefully: in an extreme scenario, i.e., when $P_{cf} =P_1$, the mixing weights reduce to $\lambda_1=0$  and $\lambda_0 =1$ causing the robust optimization to degenerate to a standard binary classification. The Wasserstein barycenters are then identical to the class-distributions and the signal of counterfactuals correctly has no influence on $Q_0^*$. If the reviewer refers to an even more extreme situation where $\epsilon_c = \delta_c = 0$ for $c \in \{0,1\}$, this would require $\mu = P_c = P_{cf}$, i.e., the class and CF samples collapse to the same distribution. This contradicts the reject-to-accept CF setting and results in a degenerated, ill-posed formulation. In particular, the feasible set becomes empty, rendering the robust optimization problem infeasible and thus ill-defined. We agree with the reviewer that discussing such extreme scenarios might help understand the formulation better: the radii capture the relative geometry between $P_0$, $P_1$, and $P_{cf}$, which is encoded via $\lambda_c$. We will make this intuition and the corresponding edge cases more explicit in the revised version.
>
> - **W3 (discussion of prototype-based structure to avoid overfitting + high fidelity):**
> Our improved generalization stems from a distribution-level, geometry-based inductive bias. Instead of fitting decision boundaries or individual samples, which are non-identifiable and prone to overfitting under one-sided CFs and low query regime, we estimate class-level prototypes via Wasserstein barycenters, which induces distributional smoothing and reduces variance. This is formally supported by Corollary A.4, which establishes stability of the induced decision score and yields a bound on prediction disagreement, directly explaining the high fidelity. Each prototype aggregates distributional information which inherently performs smoothing, averaging out sample-level noise while preserving the structure relevant to the target’s decisions. This aggregation effectively increases the usable signal per query, mitigating overfitting and enabling high-fidelity reconstruction even in low-query regimes. We will expand this discussion in the revised version.
>
> - **Q2 (sensitivity to sinkhorn regularization):** We conducted an ablation on Adult by sweeping the Sinkhorn blur from 0.2 to 0.005. Fidelity remains stable across the range of values, while the reference-set mean margin $ E_x | W_2 (x, Q_1^\star) - W_2(x, Q_0^\star) | $ decreases slightly (e.g., 82.66  to 81.08 ; query size 50). This indicates that smaller blur leads to a modest reduction in margin, while the predicted labels remain largely unchanged, signaling stable decision behavior despite changes in regularization.
> We discuss the theoretical implications as well as approximations by sinkhorn divergence in Appendix A.5 and will add an explicit reference to this section in the revised version.

---

> > ### Author Rebuttal · Reviewer_3Hmb · 2026-04-03
> >
> > Thank you for the detailed rebuttal. My concerns have been fully resolved.
> >
> > In particular, I understand how $\lambda_c$ incorporates the relative effect of $\delta_c$ and $\varepsilon_c$. Furthermore, I am convinced that the prototype structure acts as distributional smoothing and provides an inductive bias to reduce variance.
> >
> > Since my major questions were carefully addressed, I am raising my score from Weak Accept to Accept. Please ensure these clarifications are included in the final manuscript.

---

> > > ### Author Response · Authors · 2026-04-03
> > >
> > > We sincerely thank the reviewer for the engagement and for confirming that the concerns have been fully resolved.
> > > We are glad that the clarifications regarding the mixing weight and the prototype structure were convincing, and we will ensure these are incorporated into the final manuscript as suggested.
> > >
> > > We kindly note that the updated score may not yet be reflected in the system. Would the reviewer be able to verify that the intended adjustment has been registered?

---

### Official Review · Reviewer_G6uh · 2026-03-16

**Soundness:** 4
**Presentation:** 4
**Significance:** 4
**Originality:** 4
**Overall Recommendation:** 6
**Confidence:** 4

**Summary:**

The paper proposes RECAST, an attack framework that leverages one-sided counterfactual explanations (i.e., only reject-to-accept recommendations) to perform query-efficient and high-fidelity model reconstruction attacks for binary classifiers. The paper first introduces the notion of a CF-consistent feasible set $C_c$, which associates each class $c$ with the intersection between the Wasserstein ball centered at the empirical class-conditional distribution $P_c$ and the Wasserstein ball centered at the one-sided CF distribution $P_{CF}$. It then defines the CF-consistent uncertainty set in Wasserstein space as the set of joint distributions over $C_c$. RECAST then formulates the reconstruction task as prototype-based classification, in which classification is performed by comparing the Wasserstein distance of a point to distributional prototypes for the two classes. The authors evaluate the attack on four benchmark datasets, three target model classes, and five CF generation methods, and compare it to existing attacks, demonstrating its performance superiority.

**Compliance With Llm Reviewing Policy:**

Affirmed.

**Final Justification:**

Thank you for the rebuttal response. They have addressed my concerns and I have adjusted the soundness and overall score accordingly.

**Key Questions For Authors:**

1. The current experimental results only report fidelity. How good is the accuracy on $D_{ref}$? I would like some comments on the true performance of the model under attack. If the model has a high error rate, high fidelity would have a different interpretation. Therefore, it would be useful to report the accuracy of both the target model and the reconstructed models.

**Limitations:**

The limitations are, overall, appropriately discussed. One point that might benefit from a slightly more cautious framing is the discussion of auditing, which appears to assume a passive platform setting. Although the proposed solution accounts for noisy data, it has not been specifically evaluated in scenarios involving audit manipulation.

**Strengths And Weaknesses:**

### Soundness
* (+) The claims are supported by extensive empirical experiments and theoretical results.
* (+) An ablation study illustrates the importance of the different design choices.
* (+) The paper provides additional methods to characterize similarity between the original model and the surrogate (e.g., fairness diagnostics).
* (-) The datasets used are relatively small; if possible, consider including datasets such as Folktables.
* (-) The distribution shift experiments are interesting; however, they could be improved with more realistic benchmarks (e.g., Folktables with different years/states, TableShift).

### Presentation
* (+) The paper is well written.
* (+) It provides a clear motivation regarding the current limitations of CF-based attacks (decision boundary shift, unrealistic assumptions about the availability of two-sided CFs in practice, high query budgets, platform defenses such as noisy data generation, and online access during reconstruction).
* (+) The use of illustrations is helpful for building intuition and highlighting the limitations of existing attacks.

### Significance
* (+) The proposed method considers a realistic adversary model (e.g., model access, query budget, noisy data).
* (+) It achieves better performance compared to existing attacks.

### Originality
* (+) The paper contributes both theoretically, by improving the understanding of the limits of existing CF-based attacks, and methodologically, by proposing a new approach to address these limitations, supported by both extensive experiments and theoretical results.
* (+) To the best of my knowledge, the consideration of noise in this context is novel.

---

> ### Author Rebuttal · Authors · 2026-03-31
>
> Dear Reviewer,
>
> Thank you very much for acknowledging our work and contribution. We will address your remaining questions one by one below.
>
> - **W1 (small datasets):** We appreciate the concern regarding evaluation on relatively small datasets. Our method is designed for low-query regimes, where the number of accessible queries is inherently limited. In this setting, reconstruction quality is governed by query efficiency rather than dataset size, and our experiments show that RECAST achieves high fidelity well before exhausting the available samples. To further address generalizability, we additionally include results on Folktables as suggested, demonstrating consistent performance on larger-scale data. Moreover, we report runtime experiments with dimensionalities up to 784 in Appendix C.4, indicating that the method scales to higher-dimensional settings.
>
> - **W2 (distribution shifts):** Thank you for bringing this to our attention. We already included an experiment with additional noise on the adult dataset to investigate robustness under distribution shift. We agree that adding real world data reflecting distribution shifts / noise is a valuable extension which will be added for the revised version. We train a logistic-regression target model on the Folktables (ACSIncome) domain defined by California (CA), 2018 (acc=0.7860). We generate CFs using 1-nearest -neighbour and evaluate the performance (fidelity and accuracy) under distribution shift on a domain defined by Michigan (MI), 2014 on query budget (100, 150). This setup reflects a realistic cross-domain shift (both geographic and temporal).
>
> Accuracy and fidelity for different query budgets of Foktables:
> | Reconstruction | Query size = 100 |Query size = 150|
> | ----------- | ----------- |----------- |
> | | Acc. / Fid.  | Acc. / Fid.  |
> | SAMPLES | **0.7356** / 0.8641 |**0.7499** / 0.8596 |
> | CCA | 0.7302 / 0.6362 |0.7251 / 0.6251 |
> | RECAST | 0.6473 / **0.8661** | 0.7254 / **0.8886**  |
>
>
>
> Results are shown in Table above. RECAST consistently achieves higher fidelity across query budgets (100 and 150), indicating robust recovery of the target model’s behavior under distribution shift. While accuracy is lower in some cases, this is expected, as behavioral reconstruction aims to match the target model’s predictions, including its errors. Therefore, fidelity is the primary metric, and the results demonstrate that RECAST maintains strong behavioral consistency even under substantial distribution shifts.
>
> - **Q1(accuracy):** We will add this in the final version, accuracy scores are displayed in the following table. Here we perform an accuracy experiment on Adult and Compas with query size 100 and 150. The target model is logistic regression and CF method is 1-nearest-neighbor. In our setting, accuracy measures predictive performance against ground-truth labels, whereas fidelity measures agreement with the target model’s predictions; therefore these metrics capture different objectives and need not be correlated (e.g., higher-fidelity surrogates can still have lower task accuracy, and vice versa).
>
> Accuracy and fidelity on Adult and COMPAS for query sizes 100 and 150:
> | Dataset | Model | Target Acc. | Acc. (100) | Fid. (100) | Acc. (150) | Fid. (150) |
> | ----------- | ----------- |----------- |----------- | ----------- |----------- |----------- |
> | Compas |  SAMPLES   |  0.6696 |  **0.5983** |  0.7004 |  0.5424|  0.7518|
> | |  CCA  |   |  0.5405 |  0.5441 |  0.5376 |  0.5398 |
> | |  RECAST |   |  0.5580 | **0.7560** | **0.5770** |  **0.7860** |
> | Adult | SAMPLES |  0.8463 |  0.7463 |  0.7800 |  0.7488 |  0.7679 |
> | |  CCA  |  |  0.5598 |  0.5097 |  0.5318 |  0.4797 |
> | |  RECAST |   |  **0.7741** |  **0.8400** |  **0.7700** |  **0.8600** |
>
>
>
>
> - **L1 (framing of auditing):** We agree that the auditing scenario could benefit from more cautious framing. While our formulation accounts for noisy or perturbed outputs, e.g., due to privacy-preserving mechanisms, this should not be conflated with adversarial manipulation, which would require specifying explicit threat models and attack strategies. Our setting focuses on model reconstruction under limited access, where the auditor observes fixed outputs, rather than a security setting with an actively strategic platform.

---

> > ### Author Rebuttal · Reviewer_G6uh · 2026-04-02
> >
> > Thanks for the rebuttal. My concerns are fully addressed.

---

> > > ### Author Response · Authors · 2026-04-03
> > >
> > > We thank the reviewer again for the constructive feedback and are glad that the rebuttal addressed all concerns.

---

### Decision · Program_Chairs · 2026-04-30

**Decision:**

Accept (regular)

**Comment:**

Reviewers agreed that this paper presents a novel and theoretically sound framework for model reconstruction under limited data. While some initial concerns were raised regarding the impact of distribution shifts and the practical limitation to binary classification, the authors' rebuttal successfully addressed most of these points by providing additional experiments on larger-scale datasets and clarifying how to prevent decision boundary shifts.

The majority of reviewers have converged on a positive recommendation (1x Strong Accept, 2x Accept). In addition, the reviewer with Weak Reject will not insist on rejection. Therefore, the paper is recommended for acceptance to the conference.